# Characterization of the genomic landscape and actionable mutations in Chinese breast cancers by clinical sequencing

Guan-Tian Lang[1,2,8], Yi-Zhou Jiang[1,8], Jin-Xiu Shi[3,8], Fan Yang [1,2,8], Xiao-Guang Li[1], Yu-Chen Pei[1,4], Chen-Hui Zhang[3], Ding Ma[1], Yi Xiao[1], Peng-Chen Hu[3], Hai Wang[1,2], Yun-Song Yang[1,2], Lin-Wei Guo[1,2], Xun-Xi Lu[1,2], Meng-Zhu Xue[5], Peng Wang [6], A-Yong Cao[1], Hong Ling[1], Zhong-Hua Wang[1], Ke-Da Yu[1], Gen-Hong Di[1], Da-Qiang Li[1], Yun-Jin Wang[1,4], Ying Yu[7], Le-Ming Shi[7], Xin Hu[1,4✉], Wei Huang[3,4✉] & Zhi-Ming Shao [1,4✉]

The remarkable advances in next-generation sequencing technology have enabled the wide usage of sequencing as a clinical tool. To promote the advance of precision oncology for breast cancer in China, here we report a large-scale prospective clinical sequencing program using the Fudan-BC panel, and comprehensively analyze the clinical and genomic characteristics of Chinese breast cancer. The mutational landscape of 1,134 breast cancers reveals that the most significant differences between Chinese and Western patients occurred in the hormone receptor positive, human epidermal growth factor receptor 2 negative breast cancer subtype. Mutations in p53 and Hippo signaling pathways are more prevalent, and 2 mutually exclusive and 9 co-occurring patterns exist among 9 oncogenic pathways in our cohort. Further preclinical investigation partially suggests that *NF2* loss-of-function mutations can be sensitive to a Hippo-targeted strategy. We establish a public database (Fudan Portal) and a precision medicine knowledge base for data exchange and interpretation. Collectively, our study presents a leading approach to Chinese precision oncology treatment and reveals potentially actionable mutations in breast cancer.

[1] Department of Breast Surgery, Key Laboratory of Breast Cancer in Shanghai, Fudan University Shanghai Cancer Center, 270 Dong'an Road, 200032 Shanghai, P.R. China. [2] Department of Oncology, Shanghai Medical College, Fudan University, 130 Dong'an Road, 200032 Shanghai, P.R. China. [3] Department of Genetics, Shanghai-MOST Key Laboratory of Health and Disease Genomics, Chinese National Human Genome Center at Shanghai (CHGC) and Shanghai Academy of Science and Technology (SAST), 250 Bibo Road, 201203 Shanghai, P.R. China. [4] Precision Cancer Medical Center Affiliated to Fudan University Shanghai Cancer Center, 688 Hongqu Road, 201315 Shanghai, P.R. China. [5] SARI Center for Stem Cell and Nanomedicine, Shanghai Advanced Research Institute, Chinese Academy of Sciences, 201210 Shanghai, P.R. China. [6] CAS Key Laboratory of Computational Biology, CAS-MPG Partner Institute for Computational Biology, Shanghai Institute of Nutrition and Health, Shanghai Institutes for Biological Sciences, University of Chinese Academy of Sciences, Chinese Academy of Sciences, 320 Yueyang Road, 200031 Shanghai, P.R. China. [7] State Key Laboratory of Genetic Engineering, School of Life Sciences and Human Phenome Institute, Fudan University, 2005 Songhu Road, 200438 Shanghai, P.R. China. [8] These authors contributed equally: Guan-Tian Lang, Yi-Zhou Jiang, Jin-Xiu Shi, Fan Yang. ✉email: xinhu@fudan.edu.cn; huangwei@chgc.sh.cn; zhimingshao@yahoo.com

Massively parallel next-generation sequencing (NGS) has driven marked advances in our understanding of the genomic profiles of tumors[1,2]. NGS serves as a vital method for tumor diagnosis and treatment in the present era of precision oncology[3,4]. The FoundationOne CDx genomic test and the MSK-IMPACT tumor profiling assay were approved in 2017 by the U.S. Food and Drug Administration, and these approaches provide clinically feasible access to breakthrough diagnostic tracing for advanced targeted treatment and clinical trial enrollment[5–9].

Molecular subtyping and pathology-based surrogate classification provide a fundamental basis for the selection of strategies for breast cancer treatment[10]. Genomic examinations have revealed multiple potentially actionable targets in breast cancer[6,11–15]. The targeted therapeutics approved for breast cancer include multiple agents targeting HER2 (commonly referred to as human epidermal growth factor receptor 2) amplification (trastuzumab[16], pertuzumab[17], ado-trastuzumab emtansine[18], lapatinib[19], and neratinib[20], and PARP inhibitors (olaparib[21] and talazoparib)[22] for *BRCA1/2*-mutated advanced breast cancers and PI3K inhibitors (alpelisib)[23] for *PIK3CA*-mutated advanced breast cancers. Other targeted therapies, including AKT inhibitors[24,25], STAT3 inhibitors[15,26], anti-androgen therapies[27,28], etc. are currently areas of active research.

A global analysis of clinical-grade genomic and clinical data across multiple institutions could provide more effective strategies for identifying specific patient subsets, rare genomic biomarkers, and therapeutic targets. Recently, the Fudan University Shanghai Cancer Center (FUSCC) launched and developed a clinical NGS panel for the detection of somatic and germline mutations in 484 breast cancer-specific genes (Supplementary Data 1) in clinical settings. Breast cancer patients at FUSCC who agreed to be included in a study involving the detection of multiple genes were referred to the Precision Cancer Medicine Center, and tumor tissue and peripheral blood samples were collected from these patients. DNA sequencing and data analyses were subsequently conducted at the Chinese National Human Genome Center at Shanghai (CHGC). Our study constitutes a clinical prospective sequencing of breast cancer at a large scale in China and demonstrates the clinical implications and translational value for the precise treatment for breast cancer patients in China. Using mutational data from 1134 tumors, we analyze and identify the largest genomic landscape, mutation characteristics, and potential molecular targets in Chinese breast cancer.

## Results

### Description of the study scheme and gene panel of the FUSCC-BC prospective sequencing cohort.
We prospectively collected breast tumors via core needle biopsy and paired peripheral blood samples. Paired blood DNA was obtained as a control for all tumor samples, thereby filtering germline variation and identifying somatic mutations. All primary samples were collected prior to surgery, chemotherapy or endocrine therapy administration, and the metastatic samples were obtained via the biopsy of metastasis on the target organs. The tumor samples were initially pathologically verified, prepared, and transferred to CHGC for standard sequencing, and the detailed procedures are elaborated in the "Methods" section. The tumor samples were sequenced to a mean depth of coverage of 1000× and the blood samples were sequenced to a mean depth of coverage of 400×. The sample sequencing and data analysis were performed within 1 month, and the patients were subsequently informed of the testing results and referred to clinical trials if appropriate (Fig. 1a). We manually divided the enrolled breast cancer patients into three cohorts (Fig. 1b), which were locally advanced patients who referred to

neoadjuvant therapy (cohort 1), early-stage patients who referred to surgery (cohort 2), and advanced patients who referred to salvage therapy (cohort 3). We believed clinical sequencing in cohort 1 would help researchers discover predictive biomarkers and observe drug sensitivity. Moreover, though cohort 2 could not benefit from clinical sequencing in the present time, but it would help treatment decision when recurrence would happen. Complete treatment, response, and survival information in long-term follow-up will be updated on our open-access database (http://data.3steps. cn/cdataportal/study/summary?id=FUSCC_BRCA_panel_1000). Cohort 3 had chances to take precision treatment and refer to clinical trials according to sequencing results. Our study enrolled 419 neoadjuvant breast cancer patients, 606 surgical breast cancer patients, and 109 advanced breast cancer patients from April 2018 to April 2019 (Fig. 1c). We enclosed detailed clinical information of recruited patients in Supplementary Data 2. Notably, a genomically matched umbrella clinical trial (FUSCC-TNBC-Umbrella, Phase Ib/II, open-label Trial, NCT03805399)[29] was designed and conducted for the advanced triple-negative breast cancer patients using a combined index of immunohistochemistry stratification and the panel sequencing readout. This ongoing study enables Chinese clinicians to practice genomics-guided clinical treatment. Importantly, the personal information and genomic data were uploaded to the Fudan Data Portal (http://data.3steps.cn/cdataportal/study/ summary?id=FUSCC_BRCA_panel_1000) which is based on the cBioPortal open-source platform[30]. More clinical cancer genomic data and patient longitudinal follow-up data will be continuously updated in this data portal. Additionally, the Fudan Breast Cancer Precision Medicine Knowledge Base (FBC-PreMedKB, http://public-data.3steps.cn/) provides a more convenient approach for breast cancer specialists to explore actionable genomic targets and suitable drugs for clinical management[31].

We recruited more early-age breast cancer patients and more locally advanced/advanced breast cancer patients (Fig. 1d and Supplementary Table 1). Patients with III–IV stages take 40% of our cohort, while they only take 25% of the TCGA cohort. MSKCC cohort recruited more locally advanced/advanced and collected many more metastatic samples (51% versus 3%, $P <$ 0.001) comparing with ours. Moreover, 179 patients in MSKCC cohort had multiple samples from different sites sequenced, while no patient in our cohort was sequenced repeatedly (Supplementary Table 2). MSKCC cohort included more samples from diverse tissue sites like bone, pleura, and brain, which were absent in our cohort. Notably, 37% of the patients were further referred to neoadjuvant therapy, and this percentage is markedly higher than the corresponding percentage in the MSKCC and TCGA (Fig. 1d). The FUSCC panel comprised breast cancer-associated hypermutated genes in The Cancer Genome Atlas (TCGA)[32], Memorial Sloan Kettering Cancer Center (MSKCC)[33], and FUSCC-TNBC[14] datasets. Genomic biomarkers were also added to our gene panel to fulfill the future needs of basic and translational research. The genes in our panel mostly overlapped with the breast cancer-associated genes included in the FoundationOne CDx genomic test and MSK-IMPACT tumor profiling assay[6] to enable comparisons of the mutational profiles of our Chinese patients and Western patients (Supplementary Fig. 1a and Supplementary Table 3).

The patients were categorized into five subgroups, namely luminal A, luminal B/HER2−, luminal B/HER2+, HER2+, and triple-negative, based on pathological immunohistochemical staining. Our cohort comprised the largest number of primary samples among the published datasets of breast cancer, including the MSKCC and TCGA (Supplementary Fig. 1b). To ensure comparability to the stratification of the cancers included in the

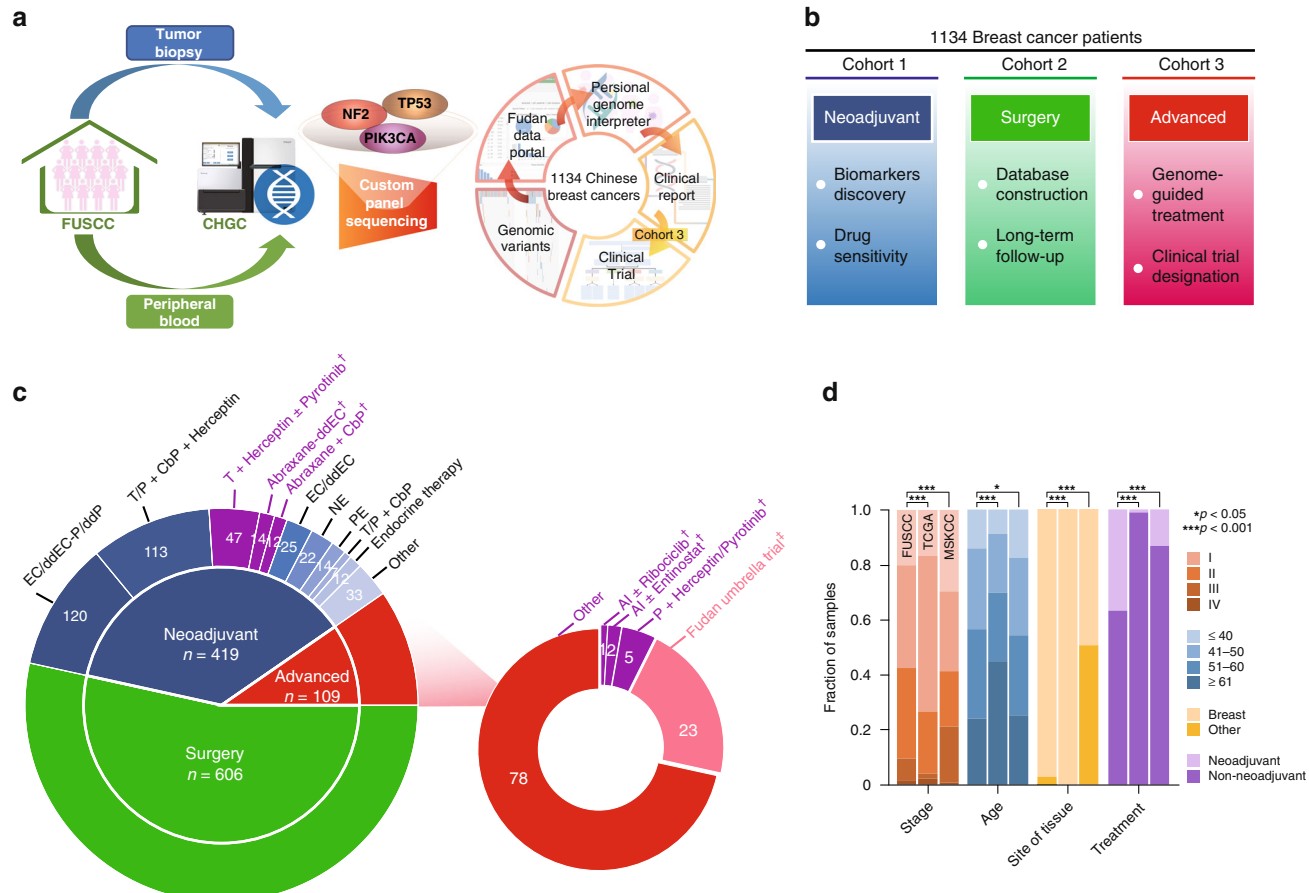

**Fig. 1 Schematic of the study and sample distributions. a** Schematic of the study. Patients in cohort 3 were referred to genome-guided clinical trials when meeting criteria. **b** Purpose of the three different investigated cohorts. **c** Treatment information. †Clinical trial. ‡Fudan Umbrella Trial. dd dose densed, EC epirubicin + cyclophosphamide, P paclitaxel, T docetaxel, NE vinorelbine + epirubicin, PE paclitaxel + epirubicin, CbP carboplatin, AI aromatase inhibitor. **d** Clinical features of our prospective cohort compared with those found in previous sequencing studies of breast cancers (MSKCC and TCGA). Source data for **d** are provided as a source data file.

MSKCC and TCGA cohorts, the breast cancer cases in our cohort were mainly grouped into the HR+/HER2− (combining luminal A and luminal B/HER2−), HR+/HER2+, HR−/HER2+, and triple-negative subtypes according to the immunohistochemical staining results. The grouping distributions are displayed in Supplementary Fig. 1c. Notably, compared with the TCGA and MSKCC cohorts, patients with HR−/HER2+, HR+/HER2+, and HR−/HER2− breast cancers were more predominant in our study cohort.

**Genomic characteristics of somatic alterations in Chinese breast cancer**. Our results illustrated the most prevalent breast cancer-related variations observed in our Chinese cohort were *TP53* mutations (53%), followed by mutations in *PIK3CA* (32%) and *NF1* (10%). The other top-ranking mutated genes are illustrated in Fig. 2a. As shown in Fig. 2b, the hotspot mutations (frequency higher than 2%) in Chinese breast cancer included *PIK3CA* p.H1047R (13%), *AKT1* p.E17K (4%), *KMT2C* p.K2797fs (2%), and *TP53* p.R248Q (2%). Recurrent mutations in *TP53* (p.H179R, p.R282W, p.R248Q, p.Y220C, p.R213X, p.R342X, p.Y163C, and p.C176Y) mainly occurred in the luminal B/HER2+, HER2+, and triple-negative subtypes of breast cancer (Supplementary Figs. 2 and 3a). *AKT1* p.E17K (9%, luminal A; 6%, luminal B/HER2−) was also enriched in the HR+/HER2− subtype. In addition, the *SF3B1* p.K700E (9%) and *SMO* p.L23del (4%) mutations were more frequent in the luminal A subtype, and

*PIK3CA* p.H1047R (16%) and *GATA3* p.P409fs (2%) were more frequently observed in the luminal B/HER2− subtype. *TP53* exhibited extremely high mutation rates in the luminal B/HER2+ (63%), HER2+ (74%), and triple-negative (80%) subtypes. Copy number variations (CNVs) were successfully determined in 1114 samples in our dataset (Fig. 2c). Frequently affected genes by CNVs in Chinese breast cancer patients were *ERBB2* (25%), *MIEN1* (25%), *GRB7* (24%), *TRPS1* (6%), *MYC* (6%), *ERLIN2* (6%), *PLPP5* (6%), *NSD3* (6%), *FGFR1* (6%), and *CCND1* (5%).

The comparisons of the variant allele frequencies (VAFs) of the top mutated genes revealed that the highest VAF ($P < 0.001$) was found for *PIK3CA* (Fig. 2d). As shown in Fig. 2e (related to Supplementary Table 4), the pairwise comparisons revealed that *AKT1* (12%) was enriched in the luminal A subtype, while *PIK3CA* (38%) and *GATA3* (13%) were enriched in the luminal B/HER2- subtype. The HER2+ subtype was significantly associated with a higher frequency of *TP53* (74%) mutations. Moreover, *TP53* (80%) and *PTEN* deficiency (10%) were distinctly enriched in TNBC. Besides these top mutated genes, we observed *FAM47C* and *KDM6A* mutations were associated with triple-negative subtype, *CBFB* mutations were associated with luminal A subtype, and *XDH* mutations were associated with luminal B/HER2- subtype.

We further investigated the mutation load in the five subtypes and found that TNBC exhibited the highest mutation load ($P = 0.005$, Supplementary Fig. 3b). Additionally, we demonstrated patterns of exclusive and co-occurring mutations (Supplementary

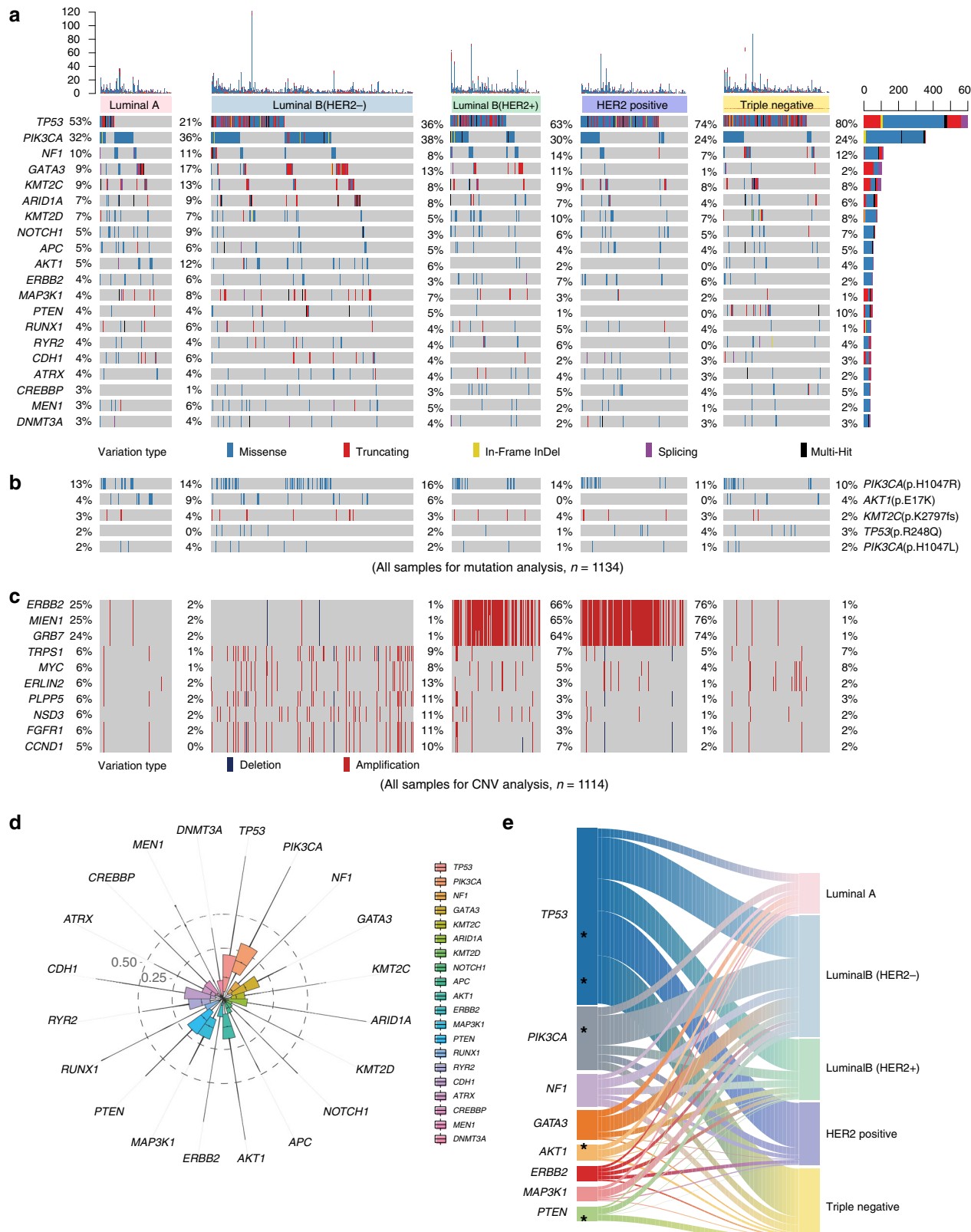

**Fig. 2 Genomic landscape and characteristics of prospectively sequenced Chinese breast cancer. a** Sequencing data of 1134 Chinese breast cancer samples classified by the molecular subtype and mutation profile and annotated with the variation type and mutation frequency. The mutation counts in each sample and each gene are provided above and on the right side, respectively. **b** Hotspot mutations (frequency higher than 2%) in Chinese breast cancer. **c** Copy number variations (CNVs) of 1114 Chinese breast cancer samples classified by the molecular subtype in our cohort. **d** Distribution of variant allele fractions (VAFs) in the recurrently mutated genes. **e** Recurrent genomic mutations (left and right) and their association with different molecular subtypes (middle). The asterisks indicate a statistically significant association with the subtype (FDR < 0.25).

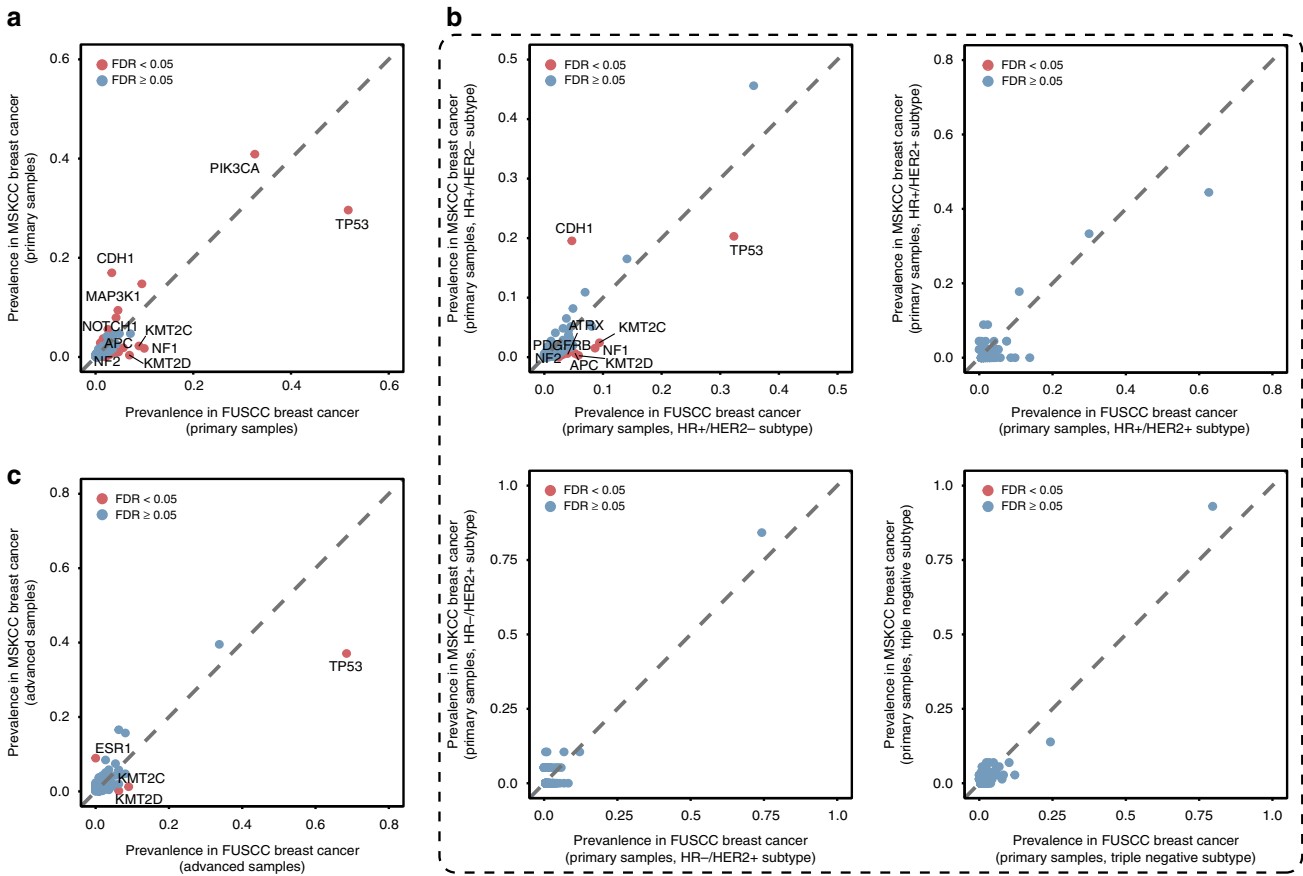

**Fig. 3 Population-specific genomic mutations in Chinese breast cancer compared with the MSKCC data. a** Scatter plots of the prevalence of mutated genes in primary breast cancer samples from the FUSCC (*x*-axis) and MSKCC (*y*-axis) datasets. **b** Scatter plots of the prevalence of mutated genes in primary breast cancer samples with the HR+/HER2− (top, left), HR+/HER2+ (top, right), HR−/HER2+ (bottom, left) and triple-negative (bottom, right) subtypes from the FUSCC (*x*-axis) and the MSKCC (*y*-axis) datasets. **c** Scatter plots of the prevalence of mutated genes in advanced breast cancer samples from the FUSCC (*x*-axis) and the MSKCC dataset (*y*-axis) datasets.

Fig. 3c, d). The highest number of co-occurring mutations was found for *NF1*, and co-occurring mutations were found in various other genes, including *RUNX1*, *APC*, *ATRX*, *KMT2D*, *MEN1*, *ARID1A*, *NOTCH1*, and *DNMT3A*. The analysis of the exclusive mutational patterns revealed that the *ARID1A* and *PTEN* mutations exhibited a completely mutually exclusive relationship (no sample harbored both *ARID1A* and *PTEN* mutations).

We investigated the differences in the mutational features between our Chinese patients and Western patients, particularly those in the published TCGA and MSKCC breast cancer datasets. Compared with the MSKCC dataset of primary breast cancer, 20 genes harbored higher and eight genes harbored lower mutation frequencies in our dataset after false discovery rate (FDR) correction (Fig. 3a). The difference was mainly concentrated in the HR+/HER2− subtype (Fig. 3b, top left), whereas only a slight difference was found in the HR+/HER2+, HR−/HER2+, and triple-negative subtypes (Fig. 3b, top right, bottom left and right), suggesting that the difference between the Chinese and Western populations is mainly found in the HR+/HER2− subtype. Moreover, we compared the advanced-stage breast cancer cases in our cohort and the MSKCC dataset and found three genes (*TP53*, *KMT2C*, and *KMT2D*) with a higher mutation frequency in our cohort and one gene (*ESR1*) with a lower mutation frequency in our dataset (Fig. 3c).

Moreover, individuals with a Caucasian ethnicity comprised 69% (757 samples) of the TCGA cohort. To comprehensively describe the differences, we compared the data of the Caucasian and African-American breast cancer cases in the TCGA with our

data of Chinese patients. A similar finding was obtained in the comparison between the Caucasian patients in the TCGA and our Chinese cohort, i.e., the largest difference between the two ethnicities occurred in the HR+/HER2− subtype (Supplementary Fig. 4a, b). Nevertheless, the mutation spectrum in our cohort was similar to that found for the African-American cohort in the TCGA (Supplementary Fig. 4c). We identified *KMT2C* p.K2797fs as a frequently mutated mutation in Chinese breast cancer but not in the MSKCC cohort (2% versus 0%, FDR < 0.05) (Supplementary Fig. 5a), and this finding was particularly true in the HR+/HER2− subtype (Supplementary Fig. 5b, top left). In addition, compared with the TCGA cohort, *AKT1* p.E17K (4% versus 0%, FDR < 0.05) and *TP53* p.R248Q (2% versus 0%, FDR < 0.05) were more frequently found in the Chinese population in our study (Supplementary Fig. 5c, top). In particular, *AKT1* p. E17K showed a higher mutation frequency (7% versus 0%, FDR < 0.05) in the HR+/HER2− breast cancer subgroup in our cohort than in the TCGA cohort (Supplementary Fig. 5d, top left).

We determined that among the 1134 patients, 5.8% (66/1134) of patients carried pathogenic/likely pathogenic germline mutations (Supplementary Fig. 6a). The mutant genes included *BRCA1* (*n* = 22), *ALK* (*n* = 5), *BRCA2* (*n* = 5), *CHEK2* (*n* = 5), *ATR* (*n* = 4), *RB1* (*n* = 4), *TP53* (*n* = 3), *TSC2* (*n* = 3), *APC* (*n* = 3), *BRAF* (*n* = 2), *NF1* (*n* = 2), *PTEN* (*n* = 2), *SMARCA4* (*n* = 2), *STK11* (*n* = 2), *ETV6* (*n* = 3), *KRAS* (*n* = 2), *MTAP* (*n* = 2), and *SMAD4* (*n* = 1).

Because sequencing results of germline variants were not directly open-access in TCGA and MSKCC database, we reviewed

published literature with germline sequencing data of large sample size. Therefore, we compared six frequently mutated genes between our Chinese cohort with published Caucasian[34] and African-American[35] cohorts. Comparing results are showed in Supplementary Fig. 6b. We found the mutation frequencies of *BRCA2* (0.44% versus 2.05% versus 1.94%, $P = 0.001$), *CHEK2* (0.44% versus 1.73% versus 0.38%, $P < 0.001$), and *PTEN* (0.18% versus 0.07% versus 0.00%, $P = 0.018$) varied among Chinese, Caucasian, and African-American population.

**Oncogenic signaling pathways in Chinese breast cancer**. We further evaluated nine canonical signaling pathways with frequent oncogenic alterations in Chinese breast cancer, including the cell cycle and the Hippo, Notch, PI-3-kinase (PI3K), β-catenin/Wnt, receptor-tyrosine (RTK)/RAS/MAP-kinase (called RTK-RAS for brevity), p53, TGF-β, and Myc signaling pathways[36]. Based on our panel sequencing results, we sketched a comprehensive diagram of mutations in these nine signaling pathways and included the mutation frequencies in each breast cancer subtype (Supplementary Fig. 7). In summary, we found that the disparity in mutational features among the different subtypes mainly occurred in the Hippo, PI3K, and p53 signaling pathways. No mutation in the Hippo signaling pathway was found in the luminal B/HER2+ subtype.

We revealed the entire landscape of pathway alterations in breast cancer, which exhibited high mutation frequencies in the p53 (55%), PI3K (45%), RTK-RAS (32%), and Notch (17%) pathways (Fig. 4a). Moreover, the mutation frequencies in the p53 (53% versus 31%, FDR < 0.05), WNT (7% versus 1%, FDR < 0.05), RTK-RAS (31% versus 18%, FDR < 0.05), Hippo (2% versus 0%, FDR < 0.05), and NOTCH (17% versus 11%, FDR < 0.05) pathways in our cohort were significantly higher in our cohort than those in the MSKCC (primary samples) (Fig. 4b). However, differences in the WNT, NOTCH, and RTK–RAS pathways were not found between the cohorts of advanced-stage samples (Supplementary Fig. 8a). Additionally, compared with the TCGA data, higher frequencies in the TGF-beta (2% versus 1%, FDR < 0.05) and Myc (1% versus 0%, FDR < 0.05) pathways were found in our Chinese population (Supplementary Fig. 8b). To be specific, incomplete genes on these pathways could limit the significance of comparisons between different sequencing cohorts. A pairwise analysis revealed that the PI3K and TGF-β signaling pathways presented higher mutation frequencies in the luminal B/HER2− subtype, the mutations in the p53 pathway were enriched in the luminal B/HER2+, HER2+, and triple-negative subtypes, and the mutations in the cell cycle occurred at a relatively higher frequency in only the HER2+ subtype (Fig. 4c). The comparison of the alterations in signaling pathways among breast cancers at different stages revealed the mutations in the p53 pathway were markedly enriched in more advanced breast cancer, e.g., stages III and IV (Supplementary Fig. 8c). Notably, the mutations in the PI3K signaling pathway were significantly enriched in the postmenopausal breast cancer patients (Supplementary Fig. 8d). The VAF patterns were diversely distributed among the signaling pathways in breast cancer. In addition, the mutation frequencies found in the Myc, p53, PI3K, and cell cycle signaling pathways were higher than those in the cell cycle, Hippo, Notch, RTK-RAS, and TGF-β signaling pathways, indicating a potential driver impact in these heavily altered pathways (Supplementary Fig. 8e). We also explored the significantly co-occurring and mutually exclusive alterations in the various pathways and found two mutually exclusive and nine co-occurring patterns in these nine oncogenic signaling pathways (Fig. 4d, left). In addition, the Hippo signaling pathway showed the most co-occurring mutations with other pathways in the

entire cohort of Chinese breast cancer (Fig. 4e). Interestingly, the co-occurring patterns of the Hippo signaling pathway only existed in the Luminal B/HER2− subtype (Fig. 4d, right) but disappeared in other subtypes (Supplementary Fig. 8f–i).

**Clinical actionability of the genomic alterations in Chinese breast cancer**. A Fudan Breast Cancer Precision Medicine Knowledge Base (FBC-PreMedKB, http://public-data.3steps.cn/) was developed[31] and utilized for our data interpretation and clinical reports. The original information regarding the actionable alterations and corresponding drugs was collected from public databases including MyCancerGenome, CIViC, Oncokb, and ClinicalTrials.gov, and optimized by our breast cancer specialists. Based on the FUSCC-BC panel, total 32 genes could be matched to 161 types of drugs (Supplementary Data 3).

We further explored potentially actionable targets in Chinese patients with breast cancer, who might benefit from clinical sequencing. The FBC-PreMedKB system was used to stratify levels of genomic biomarkers (Fig. 5a and Supplementary Fig. 9). Detailed information indicating the individual patient carrying actionable mutations and corresponding drugs is listed in Supplementary Data 4. When comparing the level 1–4 biomarkers, different levels of genomic biomarkers possessed varied components among the five molecular subtypes of breast cancer. The level 1 genomic biomarkers accounted for most actionable mutations, comprising up to 24.9% of all mutations detected in breast cancer patients (Fig. 5b). The triple-negative cases tended to have the most oncogenic mutations, and HR+/HER2− cases tended to have actionable targets among different subtypes in our cohort (Fig. 5c, d). We further generated a diagram to distinguish the features of actionability among the different subtypes (Supplementary Fig. 10a). While rare, we noticed clinically significant mutations in *CDKN2A*, which belongs to the cell cycle pathway, in the HER2+ subtype of breast cancer ($P < 0.05$), indicating the potential benefit of CDK4/6 inhibitors. The *AKT1* and *PIK3CA* actionable oncogenic mutations were significantly enriched in the Luminal A and Luminal B/HER2− subtypes, respectively ($P < 0.001$). Furthermore, we investigated the differences in the actionable features between Chinese patients and Western patients (Supplementary Fig. 10b–e). Compared with the Chinese patients, the frequency of actionable *PIK3CA* mutations was increased in both primary and advanced breast cancers in the Western patients in the MSKCC dataset according to the OncoKB criteria. Notably, the Chinese breast cancer patients harbored a higher frequency of actionable *AKT1* mutations than the Caucasian breast cancer patients. We further summarized the genomic characteristics and potential treatment strategies in the different subtypes of Chinese breast cancer studied here (Supplementary Table 5).

***NF2* loss-of-function mutations promote sensitivity to a YAP inhibitor**. Genomic alterations in the Hippo pathway were extremely rare in Caucasian breast cancer but were found at a significantly higher frequency in the Chinese population (Fig. 4b and Supplementary Fig. 7). Moreover, the Hippo pathway was found to be mostly co-altered with other pathways (Fig. 4e), providing insight into its functional synergies and potential genome-based strategies. *NF2* is a known tumor-suppressing gene that acts as a guardian in the Hippo signaling pathway[37,38]. We also found that 2% of breast cancer patients harbor *NF2* mutations, which specifically occurred in the luminal type and triple-negative breast cancer, and this frequency was higher than that found in the Caucasian population. Subsequently, we explored the role of several identified *NF2* hotspot mutations (Fig. 6a) in the signaling pathway and their responses to the YAP inhibitor

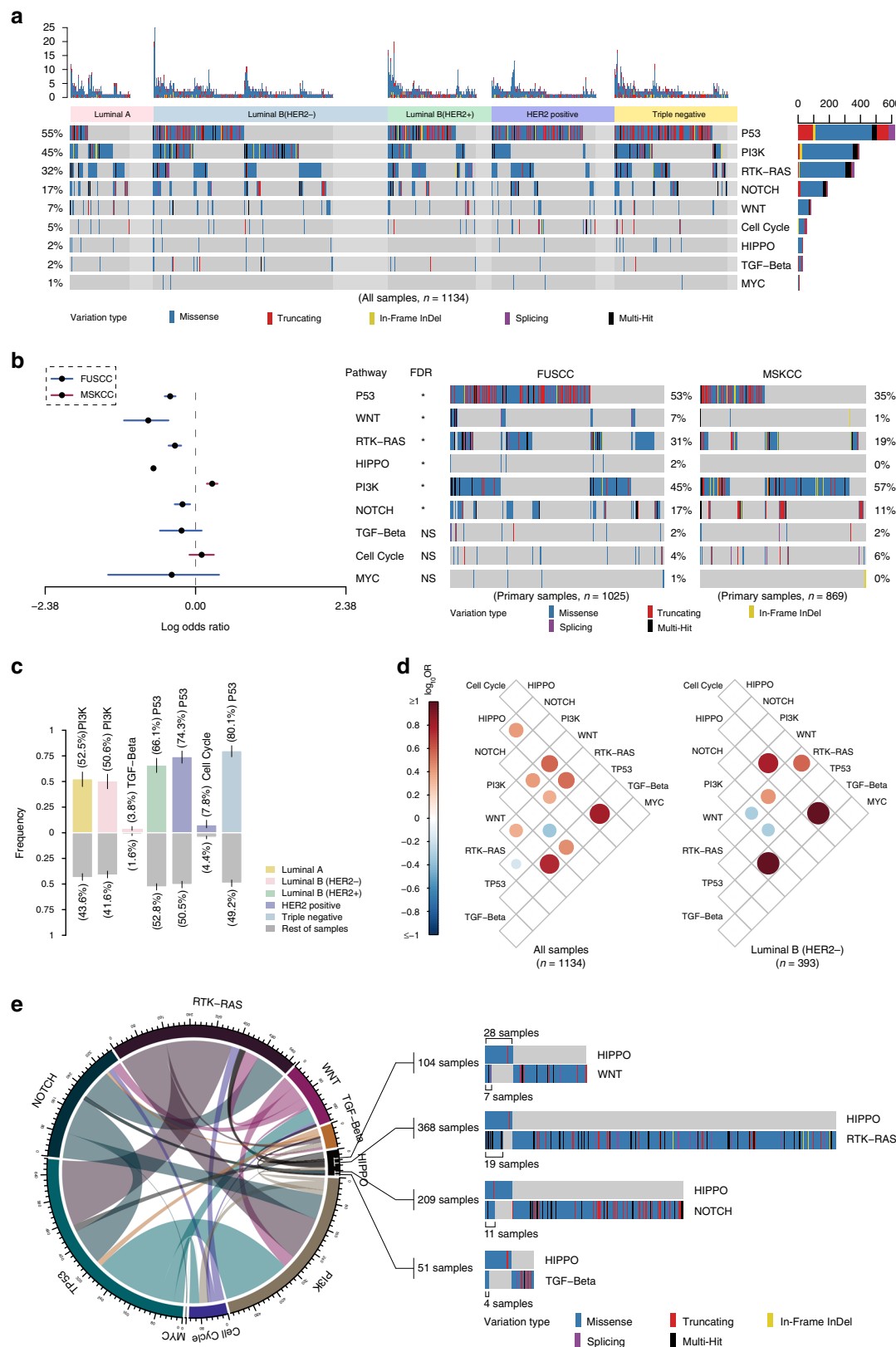

Verteporfin, which is a small molecular drug that inhibits the TEAD–YAP association[39]. *NF2* L75I, G240W, P257T/Q, and Q324K were identified as recurrent spots for they were found to be mutated in at least two cases in our cohort.

We first generated stable cell lines carrying control, wild type, or mutated NF2. We observed that the wild-type NF2 suppressed

the expression of FGFR2, while the NF2 mutants G240W and Q324K did not (Fig. 6b). The levels of phosphorylated JNK and p38, and the expression of YAP1 and cyclin D1 in the MDA-MB-231 and Hs578T cells expressing the NF2 mutants G240W and Q324K were markedly up-regulated. The exogenous wild-type NF2 suppressed the activities of Hippo and cell cycle signaling,

**Fig. 4 Characteristics of mutations in oncogenic signaling pathways in prospectively sequenced Chinese breast cancer. a** Landscape of pathway mutations in 1134 Chinese breast cancer samples classified by the molecular subtype and annotated with the variation type. The mutation counts in each sample and each pathway are provided above and on the right side, respectively. **b** Comparison of mutations in oncogenic signaling pathways in primary samples between our cohort and the MSKCC dataset. The middle circular spot corresponds to a value for odds ratio and the lines represent 95% confidence intervals. A red or blue horizontal line represents the significant or non-significant result of the comparison of mutation frequencies between our cohort and the MSKCC's cohort in each signaling pathway, respectively. The red line indicates a higher mutation frequency in the corresponding pathway favors in the MSKCC's cohort, while the blue line indicates a higher mutation frequency in the corresponding pathway favors in our cohort. A total of 1025 primary breast cancer samples from FUSCC are compared with 869 primary breast cancer samples from MSKCC by different mutation status in oncogenic signaling pathways using Fisher's exact test, adjusted by false discovery rate (FDR). The asterisks indicate FDR < 0.05. **c** Significant enrichment of pathway mutations in different molecular subtypes of breast cancer. **d** Significant mutual exclusivity (blue) and co-occurrence (red) of gene mutations in pathways in Chinese breast cancer (all samples, right; luminal B/HER2−, left). Spectrum bar: log10 (odds ratio (OR)); the color intensity represents the scale of the value. **e** Circus plot displaying the co-occurrent patterns among the oncogenic signaling pathways in our cohort. The line thickness corresponds to the number of mutations in two co-occurrent pathways. The significant co-occurrent patterns of mutations in the Hippo pathways are illustrated. Source data for **b** are provided as a source data file.

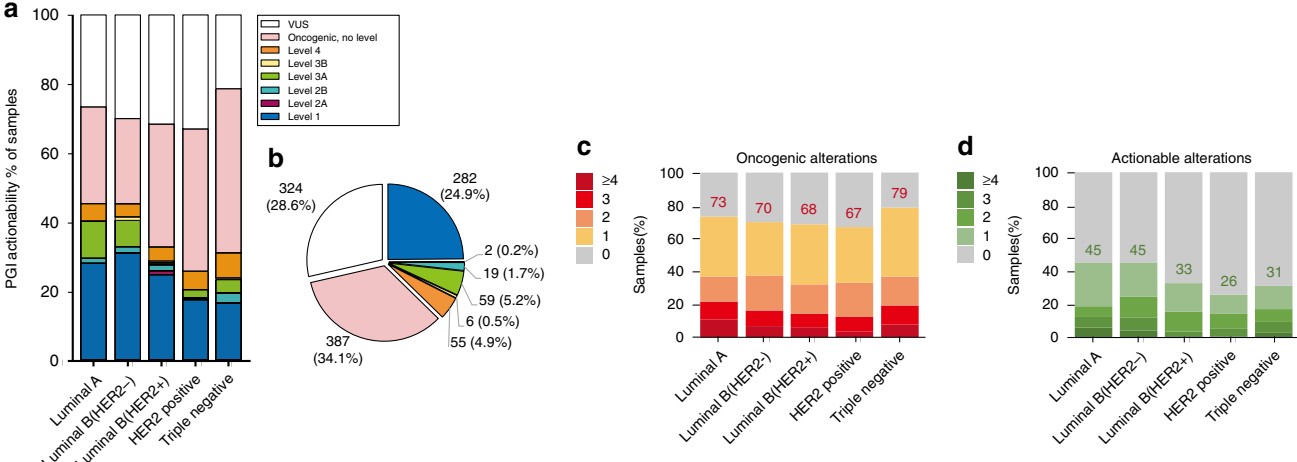

**Fig. 5 Actionable and oncogenic alterations revealed by clinical sequencing. a** Fractions of alterations annotated based on their clinical actionability according to PGI in different molecular subtypes of breast cancer. **b** Distribution of breast cancer samples assigned with the level of the most significant alteration. **c** Fractions of samples with multiple oncogenic alterations annotated in different molecular subtypes of breast cancer. **d** Actionable alterations annotated in different molecular subtypes of breast cancer.

while the loss-of-function NF2 mutants led to a disfunction of these signaling pathways. Compared with wild-type NF2, the NF2 loss-of-function mutations (G240W and Q324K) were linked to an increase in cyclin D1, which led to the activation of CDK4/6 and the phosphorylation of Rb (Fig. 6b). Verteporfin disturbed the TEAD–YAP interaction and caused the YAP degradation, and we speculate that verteporfin was able to prevent the up-regulation of YAP1, resulting in NF2 dysfunction. Hs578T cells carrying NF2-WT and G240W/Q324K were cultured in starvation medium for 24 h and then treated with increasing doses (0, 1, 3.3, and 10 μm) of Verteporfin (Fig. 6c). We also treated the same cell lines with 1 μm Verteporfin for 0, 6, 12, and 24 h (Fig. 6d). The western blot analyses indicated that treatment with Verteporfin efficiently suppressed the expression of YAP1. As expected, the cells carrying NF2 G240W/Q324K (Fig. 6e, f) were significantly more sensitive to the Verteporfin treatment than the wild-type NF2 L75I (Fig. 6g) and NF2 P257T/Q (Fig. 6h), demonstrating that NF2 L75I and NF2 P257T/Q were not completely loss-of-function mutations. We further repeated the experiment in MDA-MB-231 cells (Supplementary Fig. 11a–f). The similar results indicated that the NF2 mutant G240W (Supplementary Fig. 11d) significantly promoted sensitivity to Verteporfin and NF2 that the mutant Q324K (Supplementary Fig. 11e) had an approaching trend. Our data suggest that YAP inhibitors might be promising for the treatment of *NF2*-mutated tumors, but should be examined in further investigations.

## Discussion

Through our collective and continuous efforts, we established the largest, population-based, genomic and clinical-pathological dataset of breast cancer in China. We initially designed, created, and optimized a breast cancer-specific panel that can be applied with the NGS platform to fulfill the future needs of basic and translational research concerning clinical precision medicine. Compared with other publicly available genomic databases[6,32,33,40,41], our cohort enrolled a large number of primary breast cancer patients and mainly comprises a Chinese population. Furthermore, compared with our cohort, HR+/HER2+, HR−/HER2+ and triple-negative breast cancer were found at smaller proportions in the MSKCC[33] and TCGA[32] datasets, and thus, our work greatly increases the amount of available sequencing data and contributes to an improved understanding of these molecular subtypes. In this population-based study, we found that the mutational spectrum of the Chinese population differs from that of the foreign populations in the MSKCC dataset, and this finding was found in both primary and advanced breast cancer. Further analysis of the different molecular subtypes revealed that the distinction mainly occurred in breast cancer of the HR+/HER2− subtype of breast cancer, whereas the other subtypes showed a similar mutation prevalence. A similar finding was also obtained in a recent study involving 304 breast cancer patients using a panel of 33 genes to reveal the greatest difference in the mutational spectrum between Chinese and Caucasian

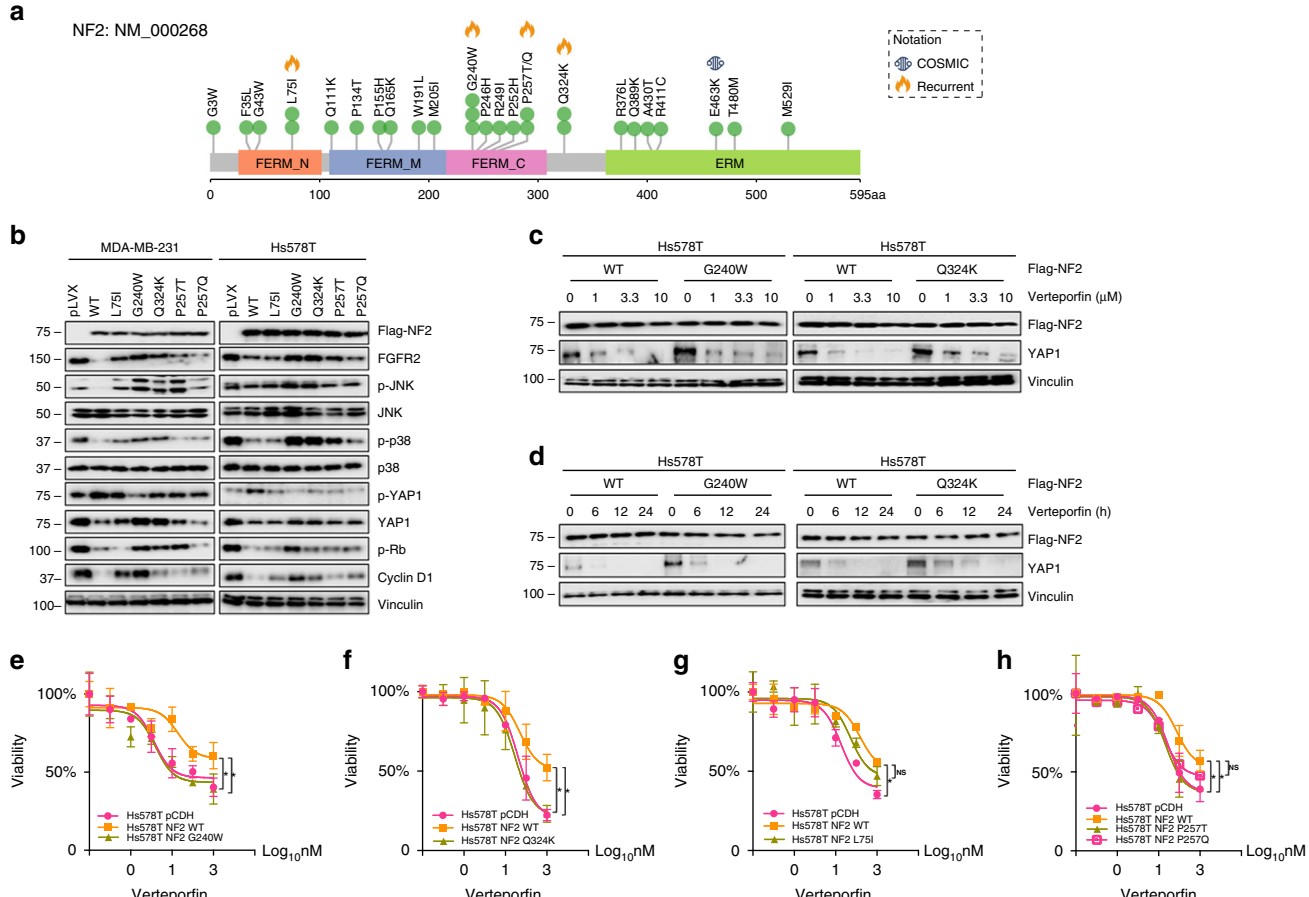

**Fig. 6 NF2 mutations promote sensitivity to a YAP inhibitor. a** *NF2* missense mutations discovered in Chinese breast cancer samples. The mutations reported in the COSMIC dataset were annotated next to the residue site. *NF2* L75I, G240W, P257T/Q, and Q324K were identified as recurrent spots in Chinese breast cancer. **b** Western blot showing the activation of Hippo and cell-cycle pathway effectors in *NF2*-wild type and *NF2*-mutated breast cancer cells. **c** Hs578T cells stably expressing Flag-*NF2*-WT and Flag-*NF2*-G240W/Q324K were treated with an increasing dose of Verteporfin. **d** Hs578T cells stably expressing Flag-*NF2*-WT and Flag-*NF2*-G240W/Q324K were treated with 1 μM Verteporfin for 0, 6, 12, and 24 h. All western blots experiments are repeated three times. **e–h** Relative percentage of cell viability (%) of *NF2*-WT and *NF2*-mutated (G240W (**e**); Q324K (**f**); L75I (**g**); P257Q/T (**h**)) Hs578T cells treated with Verteporfin. The half-maximal inhibitory concentration (IC$_{50}$) values were calculated based on the day 3 data of various doses of drug treatment. The assays were performed with five replicates in three independent experiments; representative results are shown. These data represent the mean values, with error bars indicating the SEM (*$P < 0.05$). Source data for **b**–**h** are provided as a source data file.

samples. This result was consistent with that obtained in our recent study[14]. Thus, these findings indicate that Chinese patients with triple-negative breast cancer display a mutation spectrum that is similar to that reported in foreign studies, and that *TP53* (80%), *PIK3CA* (24%), *PTEN* (10%), and *KMT2C* (8%) are some of the most frequent somatic mutations. Intriguingly, based on the given ethnic information, we found that the mutational spectrum of the Chinese dataset differs from that of the Caucasian population but is similar to that of the African-American population. We noticed two recent studies[42,43] demonstrated that *TP53, ESR1, PTEN, KMT2C, AKT1,* and *NF1* were more frequently mutated in the metastatic HR+/HER2− breast cancers compared with the early ones, in accordance with the previous study[15], indicating their driving impact in breast cancer metastasis and relapse. In our study, we found the mutation frequencies of *TP53* and *NF1* were significantly higher than those in TCGA and MSKCC datasets. First, we supposed the disparity of the mutation frequencies of *TP53* and *NF1* might result from lower proportion of patients with HR+/HER2− subtype recruited in our cohort compared to the other two cohorts. Second, the enrichment of *TP53* and *NF1* genes appears to be upheld in the metastatic HR+/HER2− subtype particularly. Third, for these

genes are enriched in metastatic breast cancer patients, it might indicate breast cancer in Chinese population is a more aggressive type compared with Caucasian population.

We further systematically explored the characteristics through multiple aspects, including VAFs, mutation enrichment, mutually exclusive and co-occurring patterns, tumor mutation counts, oncogenic pathway alterations, and clinical actionability. Mutations with high VAFs indicated early appearance during tumorigenesis or tremendous contribution to later expansion of tumor cells, and the previous study demonstrated *AKT1, CBFB, MAP2K4, ARID1A, FOXA1,* and *PIK3CA* mutations have relatively high average VAFs in breast cancers[44]. Similarly, *AKT1* and *PIK3CA* mutations harbored relatively higher VAFs in our Chinese breast cancer cohort. *PIK3CA* and *AKT1* mutations were found to be particularly enriched in the luminal B/HER2− subtype, *NF1,* and *ERBB2* mutations were enriched in the luminal B/HER2+ mutation, and *PTEN* mutations were enriched in triple-negative breast cancer. In addition, significant disparity was found among the different subtypes, suggesting a high mutation load in TNBC. Due to the comparatively moderate mutation burden of all breast cancer types[6,45], the application of immune therapy in patients with TNBC[46,47] and patients with a redundant

mutation burden is promising, as demonstrated by the somatic sequencing results[48]. The characterization of the oncogenic signaling pathways in the TCGA dataset revealed multiple features in specific molecular subtypes of breast cancer (based on PAM50 classification), whereas the samples analyzed in the previous study favored the luminal A subtype (50.8%) and a foreign population[36]. We used a Chinese population-based cohort with a larger sample size (1134) to characterize the pathway mutations in the individual molecular subtypes (based on immunostaining classification) of breast cancer. Consistent with a previous study, breast cancer was associated with high frequencies of mutations in the p53 (53%), PI3K (45%), and RTK-RAS (32%) signaling pathways.

The Hippo pathway, which was named after the *Drosophila Hpo* kinase, is a highly conserved signal transduction pathway that plays important roles in organ size control, tissue regeneration, the immune response, stem cell function, and tumor suppression[49–51]. Many studies support the role of the Hippo pathway as a tumor-suppressing pathway in diverse human cancers, including breast cancer[52,53]. Despite this finding, somatic or germline mutations in genes involved in the Hippo pathway are uncommon, and only the upstream gene *NF2* has been recognized as a bona fide tumor-suppressor gene[38]. Due to the higher frequency of alterations in the Hippo pathway and *NF2* gene in the Chinese population, we selected several mutations to verify their sensitivity to the YAP inhibitor, Verteporfin. Verteporfin was indicated for the treatment of patients with predominantly classic subfoveal choroidal neovascularization[54], and showed promising effect in cancer treatment[55–57]. We proved that *NF2* G240W and Q324K are loss-of-function mutations while L75I and P257T/Q were not. These data provide a potential genomics-based strategy for breast cancer carrying loss-of-function *NF2* mutations, and suggest that Verteporfin may exhibit higher efficacy in Hippo-activated tumors.

To be specific, for the patients receiving neoadjuvant therapy in the Cohort 1, the clinical readouts were referred to pathological complete response (pCR) and non-pCR when the patients completed the neoadjuvant treatment and underwent surgery and pathological evaluation. pCR was generally defined as complete disappearance of invasive carcinoma in the breast and regional lymph nodes, indicating drug sensitivity. In contrast, non-pCR was defined as remaining invasive carcinoma in the breast or regional lymph nodes, indicating relative resistance to neoadjuvant therapy. By comparing the mutational frequencies between the pCR and non-pCR groups in the Cohort 1, we can investigate the correlation between emerging mutations in certain genes and drug sensitivity/resistance. At present, the genomic results did not affect the clinical decisions in cohort 1. These patients received standard therapeutic treatments according to the Chinese Anti-Cancer Association guidelines for the treatment of breast cancer. Moreover, the cohort 3 included 109 advanced breast cancer patients. Detailed information of these patients is listed in Supplementary Table 1. Although most patients underwent clinical genomic sequencing, only 23 patients with refractory metastatic TNBC were ultimately enrolled in FUTURE trial (ClinicalTrials. gov identifier: NCT03805399)[29]. The purpose of this trial is to explore the clinical applicability of genomic testing and corresponding targeted therapy for refractory metastatic TNBCs. Notably, the FUTURE trial illustrated favorable outcomes and improved objective response rate (ORR) for enrolled patients.

Taken together, our study findings comprehensively reveal the characteristics of the mutations in Chinese breast cancer and thus improve our understanding of the mutational diversity among different molecular subtypes, enable the identification of potential treatment biomarkers, and provide a basis for genomic targeting strategies and clinical trials. Additionally, we constructed the Fudan Data Portal to facilitate explorations of cancer genomics data by allowing visualization and analysis across genes, samples, and clinical data in our dataset. The data portal is updatable and our ongoing study is rapidly enrolling more breast cancer patients for clinical sequencing and management.

## Methods

**Experimental model and subject details.** Patients diagnosed with malignant breast cancer who were willing to participate in the present study were prospectively recruited. In total, 1134 consecutive Chinese patients who were treated at the Department of Breast Surgery at FUSCC from 1 April 2018 to 1 April 2019 were enrolled according to the following defined criteria: (1) female patients diagnosed with unilateral breast cancer; (2) central pathological examination of tumor specimens performed by the Department of Pathology at FUSCC (the ER, PR, and HER2 status was independently confirmed by two experienced pathologists based on an immunochemical analysis and in situ hybridization; a cut-off of <1% positively stained cells was used to indicate ER/PR negativity in the immunohistochemistry testing[58], and an HER2 status was defined by an immunohistochemistry score of 0 or 1 or a lack of HER2 amplification (ratio <2.2) demonstrated by a FISH analysis according to the American Society of Clinical Oncology/College of American Pathologists guideline)[58]; and (3) sufficient frozen tissue available for further investigation. The clinicopathological characteristics included age, histological type of the tumor, tumor size, lymph node status, histological grade, therapies, and ER, PR, HER2, and Ki67 statuses. Ki67 was determined low if ≤14% and high if >14% according to the St Gallen guidelines of 2013 (ref. [59]). According to ER, PR, HER2, Ki67 status, luminal A subtype was defined by positive ER and PR, negative HER2 and low Ki67; luminal B (HER2−) subtype was defined by positive ER or PR, negative HER2 and high Ki67; luminal B (HER2+) subtype was defined by positive ER or PR, positive HER2 regardless of Ki67 status; HER2+ subtype was defined by negative ER and PR, positive HER2 regardless of Ki67 status; triple-negative subtype was defined by negative ER, PR, and HER2 regardless of Ki67 status. All tissue samples included in the present study were obtained after the research study was approved by the FUSCC Ethics Committee, and all patients provided written informed consent.

**Biospecimen collection, quality control, and processing.** The tumor and matched blood DNA were isolated from fresh frozen biopsy samples and peripheral lymphocytes using TGuide M24 (Tiangen, Beijing, China). The purity and quantity of the total DNA were estimated by measuring the absorbance at 260 nm ($A_{260}$) and 280 nm ($A_{280}$) using a NanoDrop 2000 spectrophotometer (Thermo Scientific, Wilmington, DE, USA). The extracted DNA was considered pure and suitable for future experiments if the $A_{260}/A_{280}$ ratio was within 1.6–1.9.

**Sequencing using the FUSCC-BC panel.** A custom-designed genetic panel, comprising a hybridization-capture-based assay of 484 genes that are targets of approved and experimental therapies and frequently mutated genes in breast cancer, was used in this study. The panel was designed for the detection of mutations and small insertions and deletions. In-house generated RNA baits were utilized to capture all protein-coding exons of the target genes. The RNA baits were generated from an oligo pool synthesized by Synbio Technologies (Suzhou, China). The oligo pool was amplified into double-stranded DNA. Then, a T7 promoter site was incorporated into the amplicon, and the DNA was transcribed into biotinylated RNA. The biotinylated RNA was purified, quantified, and prepared for target enrichment.

Both the tumor and matched blood samples were sequenced. At least 10 ng of each DNA sample obtained after SYBR green quantification were fragmented using a Covaris M220 and then subjected to end-repair, A-tailing and adapter ligation using a KAPA HyperPlus kit (Kapa Biosystems) according to the manufacturer's recommended protocol. Subsequently, 750 ng of prepared DNA in a volume of 3.4 μl was captured by RNA baits, and then, the captured library was purified and amplified using index primers. After quantification with a Multi-Mode Reader (BioTek), the libraries were pooled and sequenced using an Illumina HiSeq X TEN platform (Illumina Inc., San Diego, CA, USA).

Data were collected using Illumina Real Time Analysis (RTA) and assembled to fastq files using Illumina Bcl2Fastq2. The variant calling and coverage analysis of each capture region were performed using an in-house developed bioinformatics pipeline based on the general variant calling pipeline. Briefly, the high-quality reads were mapped to the hg19 version of the human reference genome (GRCh37) using the BWA aligner with the BWA-MEM algorithm and default parameters. The Genome Analysis Toolkit was used to locally realign the BAM files at intervals with mismatched indels and recalibrate the base quality scores of the reads in the BAM files. Germline variants from the blood BAM file were identified using GATK HaplotypeCaller. Somatic mutations were called from the tissue and blood BAM files using GATK4 Mutect2 with the default parameters. The VCF files were annotated using ANNOVAR. The variants and annotation results were transferred into Excel spreadsheets. To improve the specificity, a panel of normal (PON) sample filters was used to filter the expected germline variations and artifacts[14]. Each alteration identified by the pipeline was manually reviewed to ensure that no

false positives were reported. The sequencing quality statistics were obtained using SAMtools and GATK. FACETS algorithm was utilized in detecting gene-level amplification and deletion.

A modified version of the Characterization of Germline Variants (CharGer)[60], an automated scoring system according to the guidelines provided by the American College of Medical Genetics and Genomics (ACMG)[61], was used to further classify germline variants. First, as the current version of CharGer does not include minor allele frequency (MAF) data for the East Asian population, we manually filtered our candidate variants according to Genome Aggregation Database (gnomAD). Only rare variants (MAF < 0.5% in the East Asian population) were retained. Second, we manually updated the ClinVar records, and only variants known to be related to cancer were kept for PS1 and PM5 classifications. As we disabled the PM2 module, we slightly adjusted the cut-off values used for variant classification: variants with a CharGer score higher than 7 were considered likely pathogenic mutations, and variants known to be pathogenic were classified as pathogenic mutations.

The DNA sequencing dataset in our study has been deposited in the publicly accessible database from Chinese Academy of Sciences. All data can be viewed in NODE (http://www.biosino.org/node) by pasting the accession (OEP001027) into the text search box or through the URL http://www.biosino.org/node/project/detail/OEP001027.

**Cell lines and culture**. The human embryonic kidney HEK293T (293T), MDA-MB-231, and Hs578T cell lines were obtained from the Shanghai Cell Bank Type Culture Collection Committee (CBTCCC, Shanghai, China) in 2014. The cells were cultured in DMEM (Gibco, Gaithersburg, MD, USA) supplemented with 10% fetal bovine serum (FBS) (Gibco) and 1% penicillin/streptomycin (Invitrogen, Carlsbad, CA, USA). The identities of the cell lines were confirmed by the Shanghai Cell Bank Type Culture Collection Committee (CBTCCC, Shanghai, China) using DNA profiling (short tandem repeat, STR). The cell lines were subjected to routine cell line quality examinations (e.g., by morphology and mycoplasma testing) by HD Biosciences every 3 months. Verteporfin was obtained from Selleck Chemicals (Houston, USA).

**Plasmid construction and mutagenesis**. The Flag-NF2 expression vector was kindly provided by Prof. Fa-Xin Yu (Fudan IBS Laboratory, Shanghai). NF2 mutation was generated by site-directed mutagenesis using a QuikChange II site-directed mutagenesis kit (Agilent Technologies, Palo Alto, CA, USA). All constructs were verified by a sequence analysis (HuaGene Biotech, Shanghai, China). The detailed information of the expression constructs and the primers used for the molecular cloning is provided in Supplementary Table 6. The transient plasmid transfection was performed using Neofect DNA transfection reagent (Tengyi Biotech, Shanghai, China) according to the manufacturer's recommended protocol. To generate stable cell lines expressing the cDNAs, HEK293T cells were transfected with each lentivirus expression vector and packaging plasmid mix using Neofect DNA transfection reagents. The supernatant containing the virus was collected 48 h after transfection, filtered, and used to infect the target cells in the presence of 8 μg/ml polybrene prior to the drug selection with 2 μg/ml puromycin for 1 week.

**Western blot analysis**. Whole-cell lysates were resolved using T-PER Tissue Extraction Reagent (Thermo Fisher Scientific Inc., MA, USA) with a complete ethylenediaminetetraacetic acid-free protease inhibitor and phosphatase inhibitor cocktail (Selleck Chemicals). Immunoblotting was performed using a standard method. The detailed information of the antibodies used in this study is summarized in Supplementary Table 7. The quality of the gel loading and the transfer processes were assessed by immunostaining the blots with the Vinculin antibody.

**IC$_{50}$ assays, proliferation assays, and drug responses**. For the IC$_{50}$ assays, $6 \times 10^3$ cells in the logarithmic growth stage were plated in 96-well plates. Verteporfin was purchased from Selleck Chemicals. Cells were allowed to adhere overnight, and then the medium was replaced with medium containing serially diluted concentrations of Verteporfin for 3 days. Ten microliters of CCK-8 solution (Yeason, 40203ES60) were added to each well, the plates were incubated for 3 h, and absorbance at 450 nm was determined. The IC$_{50}$ was calculated using GraphPad Prism (GraphPad Software, Inc.).

**Quantification and statistical analysis**. The data distribution was characterized by frequency tabulation and summary statistics. Student's $t$-test, an analysis of variance, the Mann–Whitney Wilcoxon test, and the Kruskal–Wallis test were utilized to compare the continuous variables and ordered categorical variables, whereas Pearson's chi-square test and Fisher's exact test were employed for the comparisons of the unordered categorical variables. The $p$ values were adjusted to the FDR using the Benjamini–Hochberg procedure for multiple comparisons. All analyses were performed using R package version 3.4.2 (https://cran.r-project.org/). The assessment of the clinical actionability of the mutations detected by MSK-IMPACT was processed according to OncoKB[62] (http://oncokb.org). Accordingly, the mutations were classified into the following separate levels: FDA-recognized biomarkers (Level 1) predict response to standard-of-care therapies (Level 2), predict response to investigational agents in clinical trials (Level 3), and oncogenic biomarkers with unidentified druggable targets. Levels 2 and 3 were subdivided

according to whether evidence exists in the pertinent tumor type (2A, 3A) or a different tumor type (2B, 3B).

**Reporting summary**. Further information on research design is available in the Nature Research Reporting Summary linked to this article.

## Data availability
The DNA sequencing dataset in our study is deposited in the publicly accessible database from the Chinese Academy of Sciences. All data can be viewed in The National Omics Data Encyclopedia (NODE) (http://www.biosino.org/node) by pasting the accession number (OEP001027) into the text search box or through the following URL: http://www.biosino.org/node/project/detail/OEP001027. The FUSCC-BC sequence data have also been deposited in the NCBI Sequence Read Archive (SRA) database under the accession code SRP282257, SRP282270 and SRP282290. All data can be viewed in the SRA website (https://www.ncbi.nlm.nih.gov/sra) by pasting the accession number (SRP282257, SRP282270, and SRP282290) into the text search box or through the following hyperlinks: https://trace.ncbi.nlm.nih.gov/Traces/study/?acc=SRP282257, https://trace.ncbi.nlm.nih.gov/Traces/study/?acc=SRP282270, and https://trace.ncbi.nlm.nih.gov/Traces/study/?acc=SRP282290. The public data that support the findings of this study are available from cBioPortal (http://download.cbioportal.org/breast_msk_2018.tar.gz, http://download.cbioportal.org/brca_tcga.tar.gz) and from Fudan Data Portal (http://data.3steps.cn/cdataportal/study/summary?id=FUSCC_BRCA_panel_1000). All the other data supporting the findings of this study are available within the article and its supplementary information files and from the corresponding author upon reasonable request. A reporting summary for this article is available as a Supplementary Information file. Source data are provided with this paper.

## Code availability
All codes and scripts are available on https://github.com/ninnywolf/FUSCC-PCMC-BRCA-target-sequencing.git. Source data are provided with this paper.

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

## Acknowledgements

This work was supported by grants from the National Natural Science Foundation of China (81572583, 81502278, 81672601, 81872137, and 81602311), the National Science and Technology Major Project (2020ZX09201-013), the Shanghai Municipal Science and Technology Project (18JC1420100, 15411953300), the National Key R&D Project of China (2017YFC0907503), the Municipal Project for Developing Emerging and Frontier Technology in Shanghai Hospitals (SHDC12010116), the Cooperation Project of Conquering Major Diseases in Shanghai Municipality Health System (2013ZYJB0302), the Innovation Team of Ministry of Education (IRT1223), the Shanghai Key Laboratory of Breast Cancer (12DZ2260100), and the Ministry of Science and Technology of China (2018YFE020160) and Shanghai leading talent training program 2017. The funders had no role in the study design, collection and analysis of the data, decision to publish, or manuscript preparation. We thank all the patients who enrolled in our study, donated tissues, and the associated pseudo-anonymized clinical data for this project.

## Author contributions

Z.-M.S., W.H., and X.H. conceived the overall study and supervised the research. G.-T.L., Y.-Z.J., J.-X.S., F.Y., X.-G.L., Y.-C.P., C.-H.Z., D.M., Y.X., P.-C.H., H.W., Y.-S.Y., L.-W.G., X.-X.L., M.-Z.X., P.W., A.-Y.C., H.L., Z.-H.W., K.-D.Y., G.-H.D., D.-Q.L., Y.-J.W., Y.Y., L.-M.S., X.H., W.H., and Z.-M.S. contributed to the literature search, data collection, and data analysis. G.-T.L. and Y.-Z.J. provided the figures and drafted the manuscript, with additional input from all authors. J.-X.S. and F.Y. designed and performed the experiments. Y.-C.P., Y.-J.W., and Y.Y. helped with data interpretation and database construction. All authors approved the final manuscript.

## Competing interests

The authors declare no competing interests.
