## [Peer Review File · Nature Communications]

Reviewers' Comments:

Reviewer #1:

Remarks to the Author:

Thank you for giving me the opportunity to review the manuscript "Characterization of the Genomic Landscape and Actionable Mutations in Breast Cancer through Clinical Sequencing", by Lang, Jiang, Shi, Yang et al.

The authors report on a large scale effort of sequencing tumor of Chinese patients with breast cancer. They designed a gene panel (484 genes), enrolled patients and present in this manuscript the analyses of 1,134 tumor samples. They compared with 2 other large cohorts of breast cancer samples, MSKCC (gene panel) and TCGA (whole-exome sequencing), performed largely on tumors from patients of European ancestry.

The data are robust and the results well documented and well presented. I believe the conclusions are strong and the paper would be of interest for the scientific community. I have only a few comments on the manuscript.

The sequencing strategy is appropriate, and importantly sequence both the normal and the tumor sample, allowing the proper filtering of germline and somatic variants. However, no analyses is provided on the germline variants. It would be important to know how many patients had a predisposition variant and how it compares with the other cohorts.

One regret is the absence of copy-number variants, which would be informative and would complete the picture if the sequencing strategy was adequate to assess these variants.

Another missing piece is the clinical follow-up and survival analysis: which drug were administered to the patients and for which outcome. Few examples are given but no overview of the results.

The figures are well designed and pretty clear.

Figure 2c shows the variant allele frequencies of the top mutated genes, but is presented without interpretation or comment about its importance. Is there anything significant in this figure? Any useful comparison with the two other cohorts?

I would like to understand better the Figure 4b (and associated supplemental figures). What is represented on the left side, labelled Log odds ratio? It should be made more explicit for the readers. Similar, Figure 4c should have a label on the y-axis

Although the manuscript is very well written and very clear, I noticed a few errors or typos that would need to be corrected. For instance:

On page 7, line 7, it's unclear what the authors mean by "thereby eliminating germline interference". I suggest something along the line of "filtering germline variation and identifying somatic mutations"

On page 7, line 13-14 not sure what is "ensure detective sensitivity"

On page 7, line 17, "if appropriately" should be "if appropriate"

On page 7, line 17, "open-accessible database" should be "open-access database". The entire sentence on line 17-21 is confusing.

Reviewer #2:

Remarks to the Author:

In this manuscript, Lang and colleagues describe the results of sequencing of 1,134 breast tumors (N=1,025 primary tumors and N=109 metastatic tumors) using Fudan-BC, a 484-gene custom targeted sequencing panel. They compare the mutational profiles of this cohort to the existing TCGA and MSK-IMPACT cohorts, provide examples of how these data were used clinically, and perform functional studies to follow up on the impact of NF2 mutations, which they find to be more frequent in their cohort than in previous cohorts. Overall, this is an important cohort that will

increase the diversity of targeted sequencing data in breast cancer, where most data to date are from non-Hispanic White populations.

Major comments:

1. Given that the authors collected and sequenced normal blood samples in this cohort, it seems they should have assessed and reported copy number alterations (for example by running FACETS). Limiting themselves to single-nucleotide and small indel variants gives an incomplete picture, especially for breast cancer as a copy number-driven tumor. This is driven home by the comparison of the Myc pathway between this cohort and TCGA (1% vs 0%), when of course Myc amplification is very common in breast cancer. I would strongly encourage the authors to add copy number alterations to their analyses (as was done in the various prior targeted sequencing efforts, including MSK-IMPACT).

2. The comparison of the Fudan-BC cohort to the TCGA and MSK-IMPACT cohorts is made complicated by the different compositions of the cohorts, which result from different recruitment strategies. The neoadjuvant cohort will enrich for more aggressive disease relative to TCGA, while the predominance of early-stage recruitment should enrich for less aggressive disease relative to MSK-IMPACT (where all patients had distant recurrence, and which also intentionally focused on ER+/HER2- disease). These different recruitment strategies ultimately make these sections comparing the three cohorts very difficult to synthesize, but it would be helpful if the authors could carefully outline the differences in recruitment strategies between their cohort and the other 2, and the expected (and observed) differences in subtype, stage, and prevalence of distant recurrence distributions generated by these differences. The genes they find to be enriched in their cohort relative to others include many examples that are also more associated with metastatic (as compared to primary) ER+/HER2- breast cancer per recent studies (Bertucci Nature 2019 and Angus Nature Genetics 2019) (e.g. TP53, KMT2C, NF1), which is interesting and should be noted and discussed in the context of the recruitment strategies.

3. The section on clinical actionability needs substantial clarification. First, are the authors assessing actionability in both the early (Cohorts 1 and 2) and advanced (Cohort 3) cohorts? None of these is "actionable" in an early-stage breast cancer patient, so clarification would be helpful. Second, I am puzzled by some of the mutations listed here as actionable, like TP53 and PTEN, as I am not aware of any data suggesting these are actionable; PTEN was recently shown to be associated with resistance to PI3K inhibition (Razavi Nature Cancer 2020), if that is the approach being considered, and TP53 is notoriously challenging to target. It would be clarifying if the authors could provide a supplementary table listing the actionable mutated genes identified in the cohort, the number of times each was identified in Cohort 1 vs 2 vs 3, and the proposed drugs that target that mutation. The 2 case studies provided are helpful illustrations of how these results were used in the Umbrella trial, but it would be useful to add to the aforementioned proposed supplementary table a column indicating if mutations identified in that gene did in fact lead to a specific targeted therapy approach for one of the 109 patients on that trial, what that specific targeted therapy approach was, and what was the best response (CR, PR, SD, or PD) for each patient put on that approach (to put these 2 case studies in context). For the 2 case studies, it would be helpful if each of them was shown in the same way, for example showing the case in (e) the same as the case in (f), with all previous treatments and responses followed by the PR, rather than the change from baseline over 4 cycles only.

4. For the NF2 mutations identified in the cohort: I have a few clarifying questions:

- Can the authors clarify in Figure 6A how recurrent the 4 recurrent mutations were (like just 2 cases, or more)?

- Do existing tools like POLYPHEN-2 also suggest that G240W and Q324K are LOF while the others are not? What do these tools say about the other NF2 mutations identified in this cohort that were not functionally tested, and what about the 4 NF2 mutations that were found in TCGA or MSK-IMPACT (per cbiportal, e.g. H195P, E166Q, Q320H)?

- In Figure 4B, it states the rounded 0% for the HIPPO pathway for MSK-IMPACT; I would at least give the exact percentage in the main text (e.g. 0.02% or whatever) so as not to give the impression these are completely absent in other datasets.
- In the tumors where these NF2 mutations were found, is there evidence for loss of the wild-type allele (LOH or loss of the other chromosome)?
- Does the cohort show any examples of deep deletion of the NF2 locus separate from these mutated cases?
- Can the authors comment on the distribution of the NF2 mutations in Cohorts 1 vs 2 vs 3 and by subtype? (I.e. was there a signal that they tended to be enriched in the early vs advanced-stage cohorts?)

Minor comments:

1. The second paragraph of the introduction, discussing targeted approaches in breast cancer, should be clarified a bit. I would write something like: "Approved targeted therapeutics in breast cancer include multiple agents aimed at HER2 amplification [note that there are several approved beyond the 3 chosen to be listed in the current version], as well as PARP inhibitors for BRCA1/2-mutated advanced breast cancers and PI3K inhibitors for PIK3CA-mutated advanced breast cancers. Other targeted therapies, including AKT inhibitors, STAT3 inhibitors, anti-androgen therapies, and many others are areas of active research." I do not consider CDK4/6 inhibitors to be targeted at this time, given no evidence for differential benefit for any one biomarker (e.g. CCND1 amplification, which was tested and failed as a biomarker of response).
2. The authors state in the first section of the results and Figure 1b that the neoadjuvant cohort results were used for biomarker discovery and observation of drug sensitivity/resistance, but none of these results was reported. Please be explicit that these are long-term plans for the cohort to be reported in future manuscripts. They also state that patients were referred to clinical trials as appropriate – is this only relevant to Cohort 3? Figure 1a makes it seem that neoadjuvant and surgery cohort patients were also referred to clinical trials.
3. On page 11, the authors state that 71 genes had higher and 8 genes had lower mutation frequencies compared with MSK-IMPACT. What threshold is this based on? (It does not appear to be $FDR < 0.05$? If not, should also report number of genes meeting that threshold in both directions). What is the authors' hypothesis for this imbalance, with far more genes with higher frequencies in Fudan-BC as compared to MSK-IMPACT than the reverse? It might suggest a more uniform population of breast cancers, or a more sensitive assay.
4. Also on page 11, this sentence confuses me "AKT1 and TP53 were more frequently found in the Caucasian population in our study compared with the TCGA cohort". Should this read "Chinese", not "Caucasian"?
5. On page 5, please state how these nine signaling pathway gene sets were selected.
6. On page 15, what is meant by "refractory bilateral relapse"? For distant metastasis, "bilateral" seems an odd choice of words?
7. For Figure 1, "c" should read "trial" (typo), and abbreviations are needed for these treatments. (NE, PE, and PC are not standard abbreviations at least to my knowledge.)
8. For Figure 2: for "a", how did the authors define the 5 subtypes based on IHC? Did they use Ki67 to distinguish luminal A and B? Perhaps I missed this? For "c", perhaps go out to VAF 0.5 rather than 1.0 (it is very hard to compare these VAFs, as they are quite small, and 0.5 would represent a CCF of 1.0 for a diploid tumor with 100% purity). For "d", the Sankey plot is a little difficult to follow – could they just do stacked barplots, with one bar for each of the tumor subtypes and then stacks of each of the 8 mutations? Could set these bars all to 1.0 total (same

height), which would help us see the distribution of mutations (as they already show the breakdown by subtype in (a)).

**Point-by-point responses to the referees' comments**

**Reviewer #1 (Reviewer Comments to the Author):**

**Thank you for giving me the opportunity to review the manuscript “Characterization**
**of the Genomic Landscape and Actionable Mutations in Breast Cancer through Clinical**
**Sequencing”, by Lang, Jiang, Shi, Yang et al. The authors report on a large scale effort of**
**sequencing tumor of Chinese patients with breast cancer. They designed a gene panel (484**
**genes), enrolled patients and present in this manuscript the analyses of 1,134 tumor**
**samples. They compared with 2 other large cohorts of breast cancer samples, MSKCC**
**(gene panel) and TCGA (whole-exome sequencing), performed largely on tumors from**
**patients of European ancestry. The data are robust and the results well documented and**
**well presented. I believe the conclusions are strong and the paper would be of interest for**
**the scientific community. I have only a few comments on the manuscript.**

**Reviewer #1 Major Comments:**

**Comment 1: The sequencing strategy is appropriate, and importantly sequence both**
**the normal and the tumor sample, allowing the proper filtering of germline and somatic**
**variants. However, no analyses is provided on the germline variants. It would be**
**important to know how many patients had a predisposition variant and how it compares**
**with the other cohorts.**

**Response:** We appreciate the reviewer's kind recommendations. Sequencing normal
samples is mainly used for the proper filtering of germline variants. We deeply agree that
providing the sequencing results of germline variants could tremendously contribute to a better
understanding of breast cancer predisposition in the Chinese population. We established a
pipeline for analyzing and filtering germline variants.

A modified version of the Characterization of Germline Variants (CharGer),¹ which is an
automated scoring system according to the guidelines provided by the American College of
Medical Genetics and Genomics (ACMG),² was used to further classify the germline variants.

First, as the current version of CharGer does not include minor allele frequency (MAF) data for
the East Asian population, we manually filtered our candidate variants according to the Genome
Aggregation Database (gnomAD). Only rare variants (MAF <0.5% in the East Asian population)
were retained. Second, we manually updated the ClinVar records, and only variants known to
be related to cancer were retained for the PS1 and PM5 classifications. In addition to the above-
mentioned modifications, we ran CharGer according to the protocol in a previous publication.³
As we disabled the PM2 module, we slightly adjusted the cut-off values used for the variant
classification as follows: the variants with a CharGer score higher than 7 were considered
“likely pathogenic variants”, and the variants known to be pathogenic were classified as
“pathogenic variants”. **Please find the corresponding description of methods on Page 26,**
**Line 21-25 and Page 27, Line 1-9.**

We added **Supplementary Figure 7a** to display the germline mutational results of the
breast cancer patients in our cohort. Among the 1,134 patients (66/1134, 5.8%) with
pathogenic/likely pathogenic germline mutations, the mutant genes included *BRCA1* (n = 22),
*ALK* (n = 5), *BRCA2* (n = 5), *CHEK2* (n = 5), *ATR* (n = 4), *RBI* (n = 4), *TP53* (n = 3), *TSC2* (n
= 3), *APC* (n = 3), *BRAF* (n = 2), *NF1* (n = 2), *PTEN* (n = 2), *SMARCA4* (n = 2), *STK11* (n =
2), *ETV6* (n = 3), *KRAS* (n = 2), *MTAP* (n = 2) and *SMAD4* (n = 1).

Because sequencing results of germline variants are not directly available in the TCGA
and MSKCC databases, we reviewed the published literature with germline sequencing data of
large samples. Therefore, we compared 6 frequently mutated genes between our Chinese cohort
and published Caucasian⁴ and African-American⁵ cohorts. The comparison results are shown
in **Supplementary Figure 7b**. We found that the mutation frequencies of *BRCA2* (0.44% vs
2.05% vs 1.94%, $P = 0.001$), *CHEK2* (0.44% vs 1.73% vs 0.38%, $P < 0.001$) and *PTEN* (0.18%
vs 0.07% vs 0.00%, $P = 0.018$) varied among the Chinese, Caucasian and African-American
populations. **Please find the corresponding revision on page 13, lines 4-11.**

**Supplementary Figure 7 | Germline mutations in the FUSCC-BC cohort and comparison**
 **with Caucasian and African-American breast cancers. a.** Spectrum of germline mutations
 of breast cancers in our cohort. **b.** Comparison of 6 frequently mutated genes with Caucasian
 and African-American breast cancers.

**Comment 2: One regret is the absence of copy-number variants, which would be**
 **informative and would complete the picture if the sequencing strategy was adequate to**
 **assess these variants.**

**Response:** We appreciate the reviewer’s insight regarding the absence of copy-number
 variants. Our previous concern was that target-sequencing provides copy-number variant
 results with less accuracy. The DNA copy number variations (CNVs) of a total of 1,114 samples
 were determined through the FACETS algorithm (https://github.com/dariober/cnv_facets).⁶
 According to the calls of gene-level amplification and deletion, we added **Figure 2c** to display
 the CNV results and complete the genomic landscape of Chinese breast cancer. **Please find the**
 **corresponding revision on page 10, line 18-22.** We further assessed the concordance between
 traditional methods of HER2 amplification detection (IHC and/or FISH) and *ERBB2*
 amplification detection by our assay (**Table R1**). The concordance between *ERBB2*
 amplification by sequencing and HER2 positivity by IHC/FISH is summarized in the table
 below. The overall concordance was 89.3% (995/1114).

1 **Table R1. The concordance between HER2 amplification detected by IHC/FISH and**
 2 **ERBB2 amplification detection by our assay.**

		IHC/FISH	
		Amplified	Non-amplified
FUSCC-BC	Amplified	267	12
	Non-amplified	107	728

3 **Fig. 2 | Genomic landscape and characteristics of prospectively sequenced Chinese**
 4 **breast cancer. a.** Sequencing data of 1,134 Chinese breast cancer samples classified by the
 5 molecular subtype and mutation profile and annotated with the variation type and mutation

frequency. The mutation counts in each sample and each gene are provided above and on the
right side, respectively. **b.** Hotspot mutations (frequency higher than 2%) in Chinese breast
cancer. **c.** Copy number variations (CNVs) of 1,114 Chinese breast cancer samples classified
by the molecular subtype in our cohort. **c. Copy number variations (CNVs) of 1,114 Chinese**
**breast cancer samples classified by the molecular subtype in our cohort.** **d.** Distribution of
variant allele fractions (VAFs) in the recurrently mutated genes. **e.** Recurrent genomic
mutations (left and right) and their association with different molecular subtypes (middle). The
asterisks indicate a statistically significant association with the subtype (FDR< 0.25).

**Comment 3: Another missing piece is the clinical follow-up and survival analysis:**
**which drug were administered to the patients and for which outcome. Few examples are**
**given but no overview of the results.**

**Response:** We thank the reviewer for this comment. According to the reviewer's insightful
suggestion, we further complement the information of the clinical follow-up and survival of the
whole cohort. However, our patients were mainly recruited from April 1, 2018 to April 1, 2019,
and the insufficient follow-up might limit the significance of a survival analysis (**Figure R1;**
**Figure R1a for DFS, Figure R1b for OS**), which was not performed in our study. The long-
term follow-up data will be updated on our open-access website
(http://data.3steps.cn/cdataportal/study/summary?id=FUSCC_BRCA_panel_1000). Moreover,
the treatment information of locally advanced patients who were referred to neoadjuvant
therapy and advanced patients who were referred to salvage therapy is temporarily available.
The complete treatment and response information will also be updated on our open-access
website. We enclose the corresponding information (columns entitled with *Treatment_Regimen,*
*Diagnosis_Time, Last_Contact_Time, Vital_Status, Relapse_Time, Death_Time,*
*Follow_Up_Time, DFS_Months, DFS_Event, OS_Months and OS_Event*) in **Supplementary**
**Table 2.**

**Figure R1.** Kaplan–Meier plot of progression-free survival (DFS) and overall survival (OS) in
 our cohort. **a.** Kaplan–Meier plot of DFS in our cohort. **b.** Kaplan–Meier plot of OS in our
 cohort.

**Comment 4: Figure 2c shows the variant allele frequencies of the top mutated genes,**
 **but is presented without interpretation or comment about its importance. Is there**
 **anything significant in this figure? Any useful comparison with the two other cohorts?**

**Response:** We apologize for our negligence. The mutations with variant allele frequencies
 exhibited an early appearance during tumorigenesis or tremendously contributed to the
 subsequent expansion of tumor cells, and the previous study demonstrated that the *AKT1*, *CBFB*,
 *MAP2K4*, *ARID1A*, *FOXA1* and *PIK3CA* mutations have relatively high average VAFs in breast
 cancers.⁷ Similarly, the *AKT1* and *PIK3CA* mutations harbored relatively higher VAFs in our
 Chinese breast cancer cohort. **Please find the corresponding revision on page 21, lines 16-**
 **20.**

According to the reviewer’s insightful suggestion, we further compared the variant allele
 frequencies of the frequently mutated genes with those in two other cohorts, and the results are
 shown as follows.

**Figure R2. Comparison of the variant allele frequencies of the frequently mutated genes**
 **among the FUSCC, MSKCC and TCGA datasets.**

Most genes show higher variant allele frequencies in the Caucasian cohorts (**Figure R2**).
 However, notably, the clinical specimens obtained at our hospital were mainly from small
 biopsy specimen and, thus, generally had a lower tumor purity than those observed in the TCGA
 samples, which were collected from operation specimen for multi-dimensional profiling (DNA
 sequencing, RNA-seq, SNP arrays, DNA methylation, proteomics, etc.). Therefore, we believe
 that it may be more appropriate to analyze the variant allele frequencies in the same sequencing
 strategy within our cohort.

**Comment 5: I would like to understand better the Figure 4b (and associated**
 **supplemental figures). What is represented on the left side, labelled Log odds ratio? It**
 **should be made more explicit for the readers. Similar, Figure 4c should have a label on**
 **the y-axis.**

**Response:** We appreciate the reviewer’s comments and apologize for the confusion. On
 the left side of **Figure 4b (and the associated Supplemental Figure 8a and 8b)**, the middle

circular spot corresponds to the odds ratio value, and the lines represent the 95% confidence
intervals. A red or blue horizontal line represents a significant or non-significant result in the
comparison of the mutation frequencies between our cohort and the MSKCC's cohort in each
signaling pathway, respectively. The red line indicates a higher mutation frequency in the
corresponding pathway favoring the MSKCC cohort, while the blue line indicates a higher
mutation frequency in the corresponding pathway favoring our cohort. **Please find the**
**corresponding explanation in the legend of Figure 4 on page 45, lines 6-12.**

We apologize for not labeling the y-axis in **Figure 4c**, and we added the label "frequency"
in our revised manuscript.

**Fig. 4 | Characteristics of mutations in oncogenic signaling pathways in prospectively**
**sequenced Chinese breast cancer. a.** Landscape of pathway mutations in 1,134 Chinese
breast cancer samples classified by the molecular subtype and annotated with the variation
type. The mutation counts in each sample and each pathway are provided above and on the
right side, respectively. **b.** Comparison of mutations in oncogenic signaling pathways in
primary samples between our cohort and the MSKCC dataset. **The middle circular spot**
**corresponds to the odds ratio value, and the lines represent the 95% confidence intervals. A**
**red or blue horizontal line represents a significant or non-significant result in the comparison**
**of the mutation frequencies between our cohort and the MSKCC cohort in each signaling**
**pathway, respectively. The red line indicates a higher mutation frequency in the corresponding**
**pathway favoring the MSKCC cohort, while the blue line indicates a higher mutation**
**frequency in the corresponding pathway favoring our cohort. c.** Significant enrichment of
pathway mutations in different molecular subtypes of breast cancer. **d.** Significant mutual
exclusivity (blue) and co-occurrence (red) of gene mutations among pathways in Chinese
breast cancer (all samples, right; luminal B/HER2-, left). Spectrum bar: log₁₀ (odds ratio
(OR)); the color intensity represents the scale of the value. **e.** Circus plot displaying the co-
occurrent patterns among oncogenic signaling pathways in our cohort. The line thickness
corresponds to the number of mutations in two co-occurrent pathways. The significant co-
occurrent patterns of mutations in the Hippo pathways are illustrated.

**Reviewer #1 Minor Comments:**

**Comment 1: On page 7, line 7, it's unclear what the authors mean by “thereby**
**eliminating germline interference”. I suggest something along the line of “filtering**
**germline variation and identifying somatic mutations”.**

**Response:** We are grateful for the reviewer’s helpful suggestion. We rewrote the sentence
as follows: Paired blood DNA was obtained as a control for all tumor samples, thereby filtering
germline variation and identifying somatic mutations. **Please find the corresponding**

correction on page 7, lines 5-6.

**Comment 2: On page 7, line 13-14 not sure what is “ensure detective sensitivity”.**

**Response:** We thank the reviewer for this comment. The tumor samples were sequenced
to a mean depth of coverage of 1000× and the blood samples were sequenced to a mean depth
of coverage of 400×. **Please find the corresponding correction on page 7, lines 11-13.**

**Comment 3: On page 7, line 17, “if appropriately” should be “if appropriate”. On**
**page 7, line 17, “open-accessible database” should be “open-access database”.**

**Response:** According to the reviewer’s suggestions, we corrected these two grammatical
mistakes. **Please find the corresponding corrections on page 7, line 15 and line 24.**

**Comment 4: The entire sentence on line 17-21 is confusing: “The clinical sequencing**
**in neoadjuvant cohort helped with biomarkers discovery and drug sensitivity observation, the**
**surgery cohort mainly contributed to an open-accessible database construction and waited**
**for long-term follow-up, and the advanced patients had chances in genome-guided treatment**
**enrollment in the suitable clinical trials (Fig. 1b).”**

**Response:** We apologize for the confusion. The whole section was rewritten as follows:
“We manually divided the enrolled breast cancer patients into three cohorts, namely, locally
advanced patients who were referred to neoadjuvant therapy (cohort 1), early-stage patients
who were referred for surgery (cohort 2), and advanced patients who were referred to salvage
therapy (cohort 3). We believed that the clinical sequencing of cohort 1 could help researchers
discover predictive biomarkers and observe drug sensitivity. Moreover, although cohort 2 could
not currently benefit from clinical sequencing, such sequencing could help treatment decisions
if recurrence occurs. The complete treatment, response and survival information obtained in
long-term follow-up will be updated on our open-access database
(http://data.3steps.cn/cdataportal/study/summary?id=FUSCC_BRCA_panel_1000). Cohort 3
could obtain precision treatment or receive a referral to clinical trials according to the

1 sequencing results.” **Please find the corresponding corrections on page 7, lines 15-25 and**
2 **page 8, lines 1-2.**

3 _____

**Reviewer #2 (Reviewer Comments to the Author):**

**In this manuscript, Lang and colleagues describe the results of sequencing of 1,134**
**breast tumors (N=1,025 primary tumors and N=109 metastatic tumors) using Fudan-BC,**
**a 484-gene custom targeted sequencing panel. They compare the mutational profiles of**
**this cohort to the existing TCGA and MSK-IMPACT cohorts, provide examples of how**
**these data were used clinically, and perform functional studies to follow up on the impact**
**of *NF2* mutations, which they find to be more frequent in their cohort than in previous**
**cohorts. Overall, this is an important cohort that will increase the diversity of targeted**
**sequencing data in breast cancer, where most data to date are from non-Hispanic White**
**populations.**

**Reviewer #2 Major Comments:**

**Comment 1: Given that the authors collected and sequenced normal blood samples**
**in this cohort, it seems they should have assessed and reported copy number alterations**
**(for example by running FACETS). Limiting themselves to single-nucleotide and small**
**indel variants gives an incomplete picture, especially for breast cancer as a copy number-**
**driven tumor. This is driven home by the comparison of the *Myc* pathway between this**
**cohort and TCGA (1% vs 0%), when of course *Myc* amplification is very common in**
**breast cancer. I would strongly encourage the authors to add copy number alterations to**
**their analyses (as was done in the various prior targeted sequencing efforts, including**
**MSK-IMPACT).**

**Response:** We appreciate the reviewer's insight regarding the absence of copy-number
variants. Our previous concern was that target-sequencing provides copy-number variant
results with a lower accuracy, especially since our samples are all derived from core needle
biopsies. As the reviewer stated, the DNA copy number variations (CNVs) were determined
through the FACETS algorithm (https://github.com/dariober/cnv_facets).⁶ Eventually, the
CNVs of a total of 1,114 samples were successfully called. We added **Figure 2c** to display the
results of the copy-number variants and complete the genomic landscape of Chinese breast

cancer. Please find the corresponding revision on page 10, lines 18-22.

**Fig. 2 | Genomic landscape and characteristics of prospectively sequenced Chinese**
**breast cancer. a.** Sequencing data of 1,134 Chinese breast cancer samples classified by the
molecular subtype and mutation profile and annotated with the variation type and mutation
frequency. The mutation counts in each sample and each gene are provided above and on the
right side, respectively. **b.** Hotspot mutations (frequency higher than 2%) in Chinese breast
cancer. **c.** Copy number variations (CNVs) of 1,114 Chinese breast cancer samples classified
by the molecular subtype in our cohort.

We further assessed the concordance between traditional methods of HER2 amplification
detection (IHC and/or FISH) and *ERBB2* amplification detection by our assay (**Table R1**). The
concordance between *ERBB2* amplification by sequencing and HER2 positivity by IHC/FISH
is summarized in the table below. The overall concordance was 89.3% (995/1114), which was
rational from our perspective but worse than that of the MSK-IMPACT assay (the rate was 98%,
1778/1810 in the published data from MSKCC).⁸

Table R1. The concordance between HER2 amplification detected by IHC/FISH and ERBB2 amplification detection by our assay

		IHC/FISH	
		Amplified	Non-amplified
FUSCC-BC	Amplified	267	12
	Non-amplified	107	728

Additionally, an analysis was performed to compare the CNVs between our cohort and the
MSKCC cohort. In the samples from the primary breast cancer patients, we observed that
compared with the MSKCC cohort, the *ERBB2*, *PREX2*, *LYN* and *PARP1* genes were more
frequently amplified (**Figure R3a**, FDR < 0.05) and that the *GPS2*, *AURKB*, *TP53*, *CTCF*, *TEK*,
*CD274*, *CBFB* and *CDH1* genes were more frequently deleted (**Figure R3b**, FDR < 0.05) in
our cohort. Moreover, in the samples from the advanced breast cancer patients, compared with
the MSKCC cohort, we observed that the *ERBB2*, *PPARG* and *SMYD3* genes were more
frequently amplified (**Figure R3c**, FDR < 0.05) and that the *TEK*, *ESR1*, *MAP2K2* and *ATRX*
genes were more frequently deleted (**Figure R3d**, FDR < 0.05) in our cohort.

As the reviewer recommended, we also compared the frequency of *MYC* amplification
between our cohort and the MSKCC cohort, but no significant result (6.05% vs 6.97%, FDR
= 1 for primary samples; 6.60% vs 10%, FDR = 1 for advanced samples) was observed.

**Figure R3. Comparison of copy number variations between the FUSCC and MSKCC**
**cohorts. a.** Scatter plots of the prevalence of copy number amplifications in primary samples.
**b.** Scatter plots of the prevalence of copy number deletions in primary samples. **c.** Scatter plots
of the prevalence of copy number amplifications in advanced samples. **d.** Scatter plots of the
prevalence of copy number deletions in advanced samples.

**Comment 2: The comparison of the Fudan-BC cohort to the TCGA and MSK-**
**IMPACT cohorts is made complicated by the different compositions of the cohorts, which**
**result from different recruitment strategies. The neoadjuvant cohort will enrich for more**
**aggressive disease relative to TCGA, while the predominance of early-stage recruitment**
**should enrich for less aggressive disease relative to MSK-IMPACT (where all patients had**
**distant recurrence, and which also intentionally focused on ER+/HER2- disease). These**
**different recruitment strategies ultimately make these sections comparing the three**
**cohorts very difficult to synthesize, but it would be helpful if the authors could carefully**
**outline the differences in recruitment strategies between their cohort and the other 2, and**
**the expected (and observed) differences in subtype, stage, and prevalence of distant**
**recurrence distributions generated by these differences. The genes they find to be**
**enriched in their cohort relative to others include many examples that are also more**
**associated with metastatic (as compared to primary) ER+/HER2- breast cancer per recent**
**studies (Bertucci Nature 2019 and Angus Nature Genetics 2019) (e.g. TP53, KMT2C, NF1),**
**which is interesting and should be noted and discussed in the context of the recruitment**
**strategies.**

**Response:** We are grateful for the reviewer's insightful comments and helpful suggestions.
As the following tables show, compared with the TCGA cohort, we recruited more early-age
breast cancer patients and more locally advanced/advanced breast cancer patients. Patients with
stages III–IV constitute 40% of our cohort but only account for 25% of the TCGA cohort (**Table**
**R2**), rendering our cohort enriched with more aggressive disease. Although our cohort and the
MSKCC cohort have similar distributions in tumor stages (**Supplementary Table 3**), the
MSKCC cohort collected many more metastatic samples (971/1918, 51%) compared with our

samples (31/1134, 3%). Moreover, 179 patients in the MSKCC cohort had multiple samples
 from different sites sequenced, while no patient in our cohort was sequenced repeatedly
 (**Supplementary Table 4**). The MSKCC cohort included more samples from diverse tissue
 sites, such as bone, pleura and brain, which were absent in our cohort. According to the
 reviewer's suggestions, we outline the differences in patient characteristics between our cohort
 and the other cohorts and carefully discuss the differences in the corresponding recruitment
 strategies and prevalence of metastatic distributions in the revised manuscript. **Please find the**
 **corresponding revision on Page 8, Lines 19-25 and Page 9, Lines 1-3.**

Supplementary Table 3. The comparison of patient characteristics between FUSCC and TCGA/MSKCC cohorts.

Variables	FUSCC (n=1134)		TCGA (n=982)		MSKCC (n = 1756)		P1*	P2*
	No.	(%)	No.	(%)	No.	(%)		
Age							< 0.001	0.246
≤50 years	493	43%	298	30%	802	46%		
>50 years	641	57%	684	70%	954	54%		
Sex							0.001	0.026
Female	1134	100%	971	99%	1746	99%		
Male	0	0%	11	1%	10	1%		
LN status							0.137	< 0.001
Positive	647	57%	510	52%	755	43%		
Negative	464	41%	456	46%	745	42%		
Unknown	23	2%	16	2%	246	14%		
TNM stage							< 0.001	0.040
I-II	661	58%	715	73%	1029	59%		
III-IV	453	40%	244	25%	714	41%		
Unknown	34	2%	23	2%	13	1%		
Pathological type							< 0.001	< 0.001
IDC	658	58%	728	74%	1339	76%		
ILC	12	1%	165	17%	275	16%		
Others	57	5%	89	9%	113	6%		
Unknown	407	36%	0	0%	29	2%		
ER status							< 0.001	< 0.001
Positive	720	63%	727	74%	1244	71%		
Negative	414	37%	210	21%	472	27%		
Unknown	0	0%	45	5%	40	2%		
PR status							< 0.001	< 0.001
Positive	594	52%	631	64%	1244	71%		
Negative	540	48%	305	31%	472	27%		

Unknown	0	0%	45	5%	40	2%		
HER2 status							< 0.001	< 0.001
Positive	516	46%	168	17%	190	11%		
Negative	603	53%	645	66%	1462	83%		
Unknown	15	1%	169	17%	104	6%		
Histologic grade							NA	< 0.001
I	11	1%	0	0%	90	5%		
II	310	27%	0	0%	422	24%		
III	322	28%	0	0%	1071	61%		
Unknown	491	43%	982	100%	173	10%		

*Associations were evaluated by Chi-square test. *P1*, FUSCC vs TCGA; *P2*, FUSCC vs MSKCC.

Supplementary Table 4. The comparison of sample sites of advanced patients between FUSCC and MSKCC cohorts.

Metastatic sites	FUSCC (n = 109)		MSKCC (n = 905)		P *
	No.	(%)	No.	(%)	
Liver	9	8.3%	206	22.8%	
Bone	0	0.0%	128	14.1%	
Lymph Node	9	8.3%	126	13.9%	
Chest Wall	8	7.3%	79	8.7%	
Lung	5	4.6%	64	7.1%	
Pleura	0	0.0%	39	4.3%	
Brain	0	0.0%	33	3.6%	
Breast	78	71.6%	29	3.2%	
Skin	0	0.0%	25	2.8%	
Ovary	0	0.0%	25	2.8%	
Soft Tissue	0	0.0%	24	2.7%	
Peritoneum	0	0.0%	8	0.9%	< 0.001
Epidural Mass	0	0.0%	7	0.8%	
Bowel	0	0.0%	7	0.8%	
Bladder/Ureter	0	0.0%	5	0.6%	
Stomach	0	0.0%	4	0.4%	
Orbit	0	0.0%	4	0.4%	
Uterus	0	0.0%	2	0.2%	
Pericardium	0	0.0%	2	0.2%	
Trachea	0	0.0%	1	0.1%	
Spleen	0	0.0%	1	0.1%	
Parotid	0	0.0%	1	0.1%	
Esophagus	0	0.0%	1	0.1%	
Multiple sites	0	0.0%	84	9.3%	

1 *Associations were evaluated by Fisher's exact test.

Additionally, we carefully read the two recent studies (Bertucci Nature 2019 and Angus
Nature Genetics 2019) the reviewer suggested, and both studies were cited and discussed along
with the recruitment strategies in our revised manuscript. Both studies^{9, 10} demonstrated that
*TP53*, *ESR1*, *PTEN*, *KMT2C*, *AKT1* and *NF1* were more frequently mutated in metastatic
HR+/HER2- breast cancers than early cancers, in accordance with the previous study¹¹,
indicating their driving impact in breast cancer metastasis and relapse. In our study, we found
that the mutation frequencies of *TP53*, *AKT1* and *NF1* were significantly higher than those in
the TCGA and MSKCC datasets. First, we supposed the disparity of the mutation frequencies
of *TP53* and *NF1* might result from lower proportion of patients with HR+/HER2- subtype
recruited in our cohort compared to the other two cohorts. Second, for these genes are enriched
in metastatic breast cancer patients, it might indicate breast cancer in Chinese population is a
more aggressive type compared with Caucasian population. **Please find the corresponding**
**discussion on page 21, lines 2-12.**

**Comment 3: The section on clinical actionability needs substantial clarification. First,**
**are the authors assessing actionability in both the early (Cohorts 1 and 2) and advanced**
**(Cohort 3) cohorts? None of these is “actionable” in an early-stage breast cancer patient,**
**so clarification would be helpful. Second, I am puzzled by some of the mutations listed**
**here as actionable, like TP53 and PTEN, as I am not aware of any data suggesting these**
**are actionable; PTEN was recently shown to be associated with resistance to PI3K**
**inhibition (Razavi Nature Cancer 2020), if that is the approach being considered, and**
**TP53 is notoriously challenging to target.**

**Response:** We agree with the reviewer’s observation. In the present study, we assessed
actionability in patients in both the early (Cohorts 1 and 2) and advanced (Cohort 3) cohorts.
As the reviewer insightfully indicated, none of these is actually actionable, and none of the
early-stage breast cancer patients can currently benefit from such precision treatment. However,
we propose that these data could help these patients access better treatment if recurrence occurs.
We clarified this issue in our revised manuscript.

Additionally, we apologize for previously listing some mutations that are more likely to
be responsive biomarkers as actionable targets. Some limited evidence suggests that WEE1
inhibitors have efficacy in *TP53*-mutated solid tumors.^{12, 13} Moreover, we previously noticed
that some studies demonstrated that *PTEN*-deficient tumors might respond to pan-AKT
inhibitors and PARP inhibitors.^{14, 15, 16} However, as the reviewer noted, *PTEN* was recently
shown to be associated with resistance to PI3K inhibition.¹⁷ Considering that the present
evidence is limited and controversial, we agree that it is better to remove *TP53* and *PTEN* from
the list of actionability.

**It would be clarifying if the authors could provide a supplementary table listing the**
**actionable mutated genes identified in the cohort, the number of times each was identified**
**in Cohort 1 vs 2 vs 3, and the proposed drugs that target that mutation. The 2 case studies**
**provided are helpful illustrations of how these results were used in the Umbrella trial, but**
**it would be useful to add to the aforementioned proposed supplementary table a column**
**indicating if mutations identified in that gene did in fact lead to a specific targeted therapy**
**approach for one of the 109 patients on that trial, what that specific targeted therapy**
**approach was, and what was the best response (CR, PR, SD, or PD) for each patient put**
**on that approach (to put these 2 case studies in context). For the 2 case studies, it would**
**be helpful if each of them was shown in the same way, for example showing the case in (e)**
**the same as the case in (f), with all previous treatments and responses followed by the PR,**
**rather than the change from baseline over 4 cycles only.**

**Response:** We appreciate the reviewer's comments. We agree that providing a table listing
the actionable mutated genes identified in separate cohorts could be very helpful, and such a
table (**Supplementary Table 5**) was included in our revised manuscript. As the reviewer
suggested, four columns were integrated in **Supplementary Table 5** to indicate the mutation,
targeted therapy, best response and whether it is sequencing-guided treatment. However, the
second case shown in **Figure 5h and 5i** was sequenced in October 2019, which was beyond
our recruitment. The case is presented as an example to illustrate how clinical sequencing
helped the treatment of our patients. The sequencing result of the individual case was exclusive

of the landscape of the 1,134 breast cancer patients and was not included in **Supplementary**
 **Table 5**. In **Figure 5e**, all previous treatments and related responses of the triple-negative breast
 cancer patient are shown in the same way as shown in **Figure 5h**.

 **Fig.5 | Actionable and Oncogenic Alterations Revealed by Clinical Sequencing. a.**
 Fractions of alterations annotated based on their clinical actionability according to PGI in
 different molecular subtypes of breast cancer. **b.** Distribution of breast cancer samples assigned
 with level of the most significant alteration. **c.** Fractions of samples with multiple oncogenic
 alterations annotated in different molecular subtypes of breast cancer. **d.** Actionable alterations
 annotated in different molecular subtypes of breast cancer. **e.** Time line of disease progression
 and multiple-line treatment of the advanced TNBC patient. **f.** Line charts showing the relative

tumor size of the advanced TNBC patient based on RECIST 1.1 criteria. **g.** MRI images
showing the decreasing tumor size at indicated time points. **h.** Time line of disease progression
and multiple-line treatment of the advanced HR+/HER2- breast cancer patient. **i.** MRI images
showing the decreasing liver metastasis at indicated time points. Abbreviations: Dx, diagnosis;
PR, partial response; SD, stable disease; PD, progressive disease; EC×4 = epirubicin +
cyclophosphamide for 4 cycles; P×4 = paclitaxel for 4 cycles; P+CbP = paclitaxel + carboplatin;
OFS = ovarian function suppression; AI = aromatase inhibitor.

**Comment 4: For the NF2 mutations identified in the cohort: I have a few clarifying**
**questions:**

**- Can the authors clarify in Figure 6A how recurrent the 4 recurrent mutations were (like**
**just 2 cases, or more)?**

**Response:** We are grateful for the reviewer's intuitive comments. We apologize for not
clarifying the recurrent mutations in our manuscript. *NF2* L75I, G240W, P257T/Q and Q324K
were identified as recurrent spots as they were found to be mutated in at least two cases in our
cohort. This information is provided in the revised manuscript. **Please find the corresponding**
**revision on page 18, lines 2-4.**

**- Do existing tools like POLYPHEN-2 also suggest that G240W and Q324K are LOF**
**while the others are not? What do these tools say about the other NF2 mutations identified**
**in this cohort that were not functionally tested, and what about the 4 NF2 mutations that**
**were found in TCGA or MSK-IMPACT (per cBioportal, e.g. H195P, E166Q, Q320H)?**

**Response:** As you kindly suggested, we used Polyphen2 to perform loss-of-function
prediction. *NF2* G240W was predicted as a probably damaging mutation in the Polyphen2
result, while *NF2* Q324K was not (**Table R4**).

The detailed analysis loss-of-function prediction results of *NF2* mutations in TCGA and
MSKCC cohorts through POLYPHEN-2 and SIFT are showed as in **Table R5**.

Table R4. Loss-of-function prediction results of *NF2* mutations in FUSCC cohort.

Substitutions	POLYPHEN-2
G3W	probably damaging
F35L	possibly damaging
G43W	probably damaging
L75I	probably damaging
Q111K	probably damaging
P134T	probably damaging
P155H	possibly damaging
Q165K	benign
W191L	benign
M205I	possibly damaging
G240W	probably damaging
P246H	benign
R249I	possibly damaging
P252H	probably damaging
P257T	benign
P257Q	benign
Q324K	benign
R376L	probably damaging
Q389K	benign
A403T	benign
R411C	probably damaging
E463K	possibly damaging
T480M	possibly damaging
M529I	benign

1

Table R5. Loss-of-function prediction results of *NF2* mutations in MSKCC and TCGA cohorts.

Substitutions	Function prediction results	
	POLYPHEN-2	SIFT
Q125H	probably damaging	damaging
S156R	benign	tolerated
E166Q	probably damaging	damaging
E182Q	benign	tolerated
H195P	probably damaging	damaging
E215D	benign	tolerated
Q320H	possibly damaging	damaging
E372K	probably damaging	tolerated
E404K	benign	tolerated
K586N	benign	tolerated

2

**- In Figure 4B, it states the rounded 0% for the HIPPO pathway for MSK-IMPACT;**
**I would at least give the exact percentage in the main text (e.g. 0.02% or whatever) so as**
**not to give the impression these are completely absent in other datasets.**

**Response:** We apologize for not providing the exact mutation percentage of the HIPPO
pathway for MSK-IMPACT. However, we only included *NF2* and *CSNK1E* in the HIPPO
pathway, and no sample was mutated in these two genes in the primary cohort of MSK-IMPACT.
Furthermore, we stated in our revised manuscript that, incomplete genes on these pathways
could limit the significance of such comparisons. **This statement was provided on page 14,**
**lines 10-11.**

**- In the tumors where these NF2 mutations were found, is there evidence for loss of**
**the wild-type allele (LOH or loss of the other chromosome)?**

**Response:** Notably, we find (software) that the *NF2* M205I and P252H double mutated
(case No. 1800033) sample has a loss of the X chromosome, but no other *NF2* mutated samples
show evidence of a loss of the wild-type allele or the other chromosome.

**- Does the cohort show any examples of deep deletion of the NF2 locus separate from**
**these mutated cases?**

**Response:** Interestingly, by analyzing the copy number variations through the FACETS
algorithm, we find 7 samples with a deep deletion of the *NF2* locus, and these samples are all
separate from the 25 *NF2* mutated samples (**Figure R4**); the corresponding diagram is as
follows.

**Figure R4. The diagram of *NF2* alterations in FUSCC cohort.**

**- Can the authors comment on the distribution of the NF2 mutations in Cohorts 1 vs**
**2 vs 3 and by subtype? (I.e. was there a signal that they tended to be enriched in the early**
**vs advanced-stage cohorts?)**

**Response:** We agree with the reviewer’s intuitive comments. We found that 2.2%
 (25/1,134) of breast cancer patients harbor *NF2* mutations. We observed that the *NF2* mutation
 distributions vary in different breast cancer subtypes in our study (**Table R6**, $P = 0.025$), but
 we did not observe significantly different distributions in Cohorts 1, 2 and 3 (**Table R6**, $P =$
 0.589) as shown as follows.

Table R6. *NF2* mutation distributions in different breast cancer subtypes and cohorts in our study.

		Total patients	NF2 mutation status				P
			Carriers		Non-carriers		
			N	%	N	%	
Subtypes	Luminal A	134	5	3.7%	129	96.3%	0.025
	Luminal B (HER2-)	385	8	2.1%	377	97.9%	
	Luminal B (HER2+)	174	0	0.0%	174	100.0%	
	HER2 positive	202	4	2.0%	198	98.0%	
	Triple negative	198	8	4.0%	190	96.0%	
Cohorts	Cohort 1	410	9	2.2%	401	97.8%	0.589
	Cohort 2	594	12	2.0%	582	98.0%	
	Cohort 3	105	4	3.8%	101	96.2%	

**Reviewer #2 Minor Comments:**

**Comment 1: The second paragraph of the introduction, discussing targeted**
 **approaches in breast cancer, should be clarified a bit. I would write something like:**
 **“Approved targeted therapeutics in breast cancer include multiple agents aimed at HER2**
 **amplification [note that there are several approved beyond the 3 chosen to be listed in the**
 **current version], as well as PARP inhibitors for BRCA1/2-mutated advanced breast**
 **cancers and PI3K inhibitors for PIK3CA-mutated advanced breast cancers. Other**
 **targeted therapies, including AKT inhibitors, STAT3 inhibitors, anti-androgen therapies,**
 **and many others are areas of active research.” I do not consider CDK4/6 inhibitors to be**
 **targeted at this time, given no evidence for differential benefit for any one biomarker (e.g.**
 **CCND1 amplification, which was tested and failed as a biomarker of response).**

**Response:** We sincerely thank the reviewer for this advice and amended the statement as

suggested in the revised version as follows:

“The targeted therapeutics approved for breast cancer include multiple agents targeting
HER2 (commonly referred to as human epidermal growth factor receptor 2) amplification
(trastuzumab,¹⁸ pertuzumab,¹⁹ ado-trastuzumab emtansine,²⁰ lapatinib,²¹ and neratinib),²²
PARP inhibitors (olaparib²³ and talazoparib)²⁴ for *BRCAl/2*-mutated advanced breast cancers
and PI3K inhibitors (alpelisib)²⁵ for *PIK3CA*-mutated advanced breast cancers. Other targeted
therapies, including AKT inhibitors,^{26,27} STAT3 inhibitors,^{11,28} anti-androgen therapies,^{28,29} etc.,
are currently areas of active research.” **Please find the corresponding revision on page 5,**
**lines 13-20.**

**Comment 2: The authors state in the first section of the results and Figure 1b that**
**the neoadjuvant cohort results were used for biomarker discovery and observation of**
**drug sensitivity/resistance, but none of these results was reported. Please be explicit that**
**these are long-term plans for the cohort to be reported in future manuscripts. They also**
**state that patients were referred to clinical trials as appropriate – is this only relevant to**
**Cohort 3? Figure 1a makes it seem that neoadjuvant and surgery cohort patients were**
**also referred to clinical trials.**

**Response:** We appreciate the reviewer’s comment and amended the statement that further
studies will report biomarker discovery and observations of drug sensitivity/resistance in the
neoadjuvant cohort and survival outcomes in the surgery cohort as suggested in the revised
version as follows:

“We manually divided the enrolled breast cancer patients into three cohorts, namely,
locally advanced patients who were referred to neoadjuvant therapy (cohort 1), early-stage
patients who were referred for surgery (cohort 2), and advanced patients who were referred to
salvage therapy (cohort 3). We believed that the clinical sequencing of cohort 1 could help
researchers discover predictive biomarkers and observe drug sensitivity. Moreover, although
cohort 2 could not currently benefit from clinical sequencing, such sequencing could help
treatment decisions if recurrence occurs. The complete treatment, response and survival

information obtained in long-term follow-up will be updated on our open-access database
 (http://data.3steps.cn/cdataportal/study/summary?id=FUSCC_BRCA_panel_1000). Cohort 3
 could obtain precision treatment or receive a referral to clinical trials according to the
 sequencing results.” **Please find the corresponding revision on page 7, lines 11-13.**

Regarding **Figure 1a**, we agree that this figure should show only the patients in Cohort 3
 referred to clinical trials according to their sequencing results, and we altered **Figure 1a** and
 **the corresponding figure legend** accordingly.

 **Fig. 1 | Schematic of the study and sample distributions.** **a.** Schematic of the study. **The**
 **patients in cohort 3 were referred to genome-guided clinical trials when the criteria were met.**
 **b.** Purpose of investigating three different cohorts **c.** Treatment information. † Clinical trials. ‡
 Fudan Umbrella Trial. Abbreviations: dd = dose densed, EC = epirubicin + cyclophosphamide,
 P = paclitaxel, T = docetaxel, NE = vinorelbine + epirubicin, PE = paclitaxel + epirubicin, CbP
 = carboplatin, AI = aromatase inhibitor **d.** Clinical features of our prospective cohort compared
 with those found in previous sequencing studies of breast cancers (MSKCC and TCGA).

**Comment 3: On page 11, the authors state that 71 genes had higher and 8 genes had**
**lower mutation frequencies compared with MSK-IMPACT. What threshold is this based**
**on? (It does not appear to be FDR < 0.05? If not, should also report number of genes**
**meeting that threshold in both directions). What is the authors' hypothesis for this**
**imbalance, with far more genes with higher frequencies in Fudan-BC as compared to**
**MSK-IMPACT than the reverse? It might suggest a more uniform population of breast**
**cancers, or a more sensitive assay.**

**Response:** We are grateful for the reviewer's insightful comments. We apologize for
stating that 71 genes had higher and 8 genes had lower mutation frequencies compared with
MSK-IMPACT and not providing the detailed comparison results in the previous version of our
manuscript. The previous result was stated based on a threshold of $P < 0.05$ by mistake.
Therefore, when the mutation frequencies are calculated and compared based on an $FDR < 0.05$,
we find that 20 genes had higher and 8 genes had lower mutation frequencies compared with
MSK-IMPACT. **Please find the corresponding revision on Page 11, Line 20-22.** We suppose
that the obvious imbalance disappears after the FDR correction. We provide the following table
(**Table R7**) to exhibit the detailed comparison results with the MSK-IMPACT cohort.

Table R7. Comparison of the mutation frequency between the FUSCC and MSK-IMPACT cohorts.

Gene	FUSCC Mutated Sample	MSKCC Mutated Sample	FUSCC Nonmutated Sample	MSKCC Nonmutated Sample	FUSCC Mutated Fraction	MSKCC Mutated Fraction	P	FDR
CDH1	34	137	991	670	3.3%	17.0%	< 0.001	< 0.001
TP53	530	239	495	568	51.7%	29.6%	< 0.001	< 0.001
KMT2D	71	3	954	804	6.9%	0.4%	< 0.001	< 0.001
NF1	102	14	923	793	10.0%	1.7%	< 0.001	< 0.001
KMT2C	91	18	934	789	8.9%	2.2%	< 0.001	< 0.001
APC	48	8	977	799	4.7%	1.0%	< 0.001	< 0.001
NF2	21	0	1004	807	2.0%	0.0%	< 0.001	< 0.001
NOTCH1	58	15	967	792	5.7%	1.9%	< 0.001	0.001
MAP3K1	47	76	978	731	4.6%	9.4%	< 0.001	0.001
ABL1	28	3	997	804	2.7%	0.4%	< 0.001	0.001
PIK3CA	329	330	696	477	32.1%	40.9%	< 0.001	0.002
ATRX	36	7	989	800	3.5%	0.9%	< 0.001	0.002

DNMT3A	34	6	991	801	3.3%	0.7%	< 0.001	0.002
AXIN1	18	1	1007	806	1.8%	0.1%	< 0.001	0.004
SMARCA4	24	3	1001	804	2.3%	0.4%	< 0.001	0.004
KMT2B	23	3	1002	804	2.2%	0.4%	0.001	0.006
GATA3	97	119	928	688	9.5%	14.7%	0.001	0.006
PDGFRB	22	3	1003	804	2.1%	0.4%	0.001	0.009
CBFB	26	45	999	762	2.5%	5.6%	0.001	0.009
PTEN	44	64	981	743	4.3%	7.9%	0.001	0.012
TSC2	26	5	999	802	2.5%	0.6%	0.002	0.013
ASXL1	27	6	998	801	2.6%	0.7%	0.002	0.019
CIC	25	5	1000	802	2.4%	0.6%	0.002	0.020
NOTCH3	32	9	993	798	3.1%	1.1%	0.004	0.029
BRCA2	10	23	1015	784	1.0%	2.9%	0.004	0.029
NCOR1	16	30	1009	777	1.6%	3.7%	0.004	0.029
VHL	13	1	1012	806	1.3%	0.1%	0.005	0.035
CREBBP	38	13	987	794	3.7%	1.6%	0.006	0.043

**Comment 4: Also on page 11, this sentence confuses me “AKT1 and TP53 were more**
**frequently found in the Caucasian population in our study compared with the TCGA**
**cohort”. Should this read “Chinese”, not “Caucasian”?**

**Response:** Once again, we apologize for the omission. The word “Caucasian” was
replaced with the word “Chinese”. **Please find the corresponding correction on page 12, line**
**19.**

**Comment 5: On page 5, please state how these nine signaling pathway gene sets were**
**selected.**

**Response:** We appreciate the reviewer’s comment and apologize for the confusion. The
nine signaling pathway gene sets were selected according to a study conducted by Francisco
Sanchez-Vega.²⁹ These authors analyzed the mechanisms and patterns of somatic alterations in
the following 10 canonical pathways across 33 cancer types in 9,125 samples: cell cycle, Hippo,
Myc, Notch, Nrf2, PI-3-Kinase/Akt, RTK-RAS, TGFβ, P53 and β-catenin/WNT. However,
Nrf2 signaling pathway genes were not incorporated in our FUSCC-BC panel, and we choose
the remaining nine pathways to perform our analysis.

**Comment 6: On page 15, what is meant by “refractory bilateral relapse”? For distant**
**metastasis, “bilateral” seems an odd choice of words?**

**Response:** We apologize for this mistake. This text was replaced with the phrase
“contralateral recurrence”. We revised the manuscript comprehensively to avoid such mistakes.
**Please find the corresponding correction on page 16, line 25.**

**Comment 7: For Figure 1, “c” should read “trial” (typo), and abbreviations are**
**needed for these treatments. (NE, PE, and PC are not standard abbreviations at least to**
**my knowledge.)**

**Response:** We apologize for this spelling mistake and the ambiguous illustration of the
abbreviations of the treatments. The typographical error was corrected, and the unabbreviated
forms of chemotherapy are displayed in the legend of **Figure 1**.

**“a.** Schematic of the study. The patients in cohort 3 were referred to genome-guided
clinical trials when the criteria were met. **b.** Purpose of investigating three different cohorts **c.**
Treatment information. † Clinical trials. ‡ Fudan Umbrella Trial. **Abbreviations: dd = dose**
**densed, EC = epirubicin + cyclophosphamide, P = paclitaxel, T = docetaxel, NE = vinorelbine**
**+ epirubicin, PE = paclitaxel + epirubicin, CbP = carboplatin, AI = aromatase inhibitor d.**
Clinical features of our prospective cohort compared with those found in previous sequencing
studies of breast cancers (MSKCC and TCGA).”

**Comment 8: For Figure 2: for “a”, how did the authors define the 5 subtypes based**
**on IHC? Did they use Ki67 to distinguish luminal A and B? Perhaps I missed this?**

**Response:** We are grateful for the reviewer’s helpful comments. We apologize for the
missing part of the definition of the 5 subtypes based on immunohistochemistry, and we
complement the information in the part of Methods. **Please find the corresponding revision**
**on page 24, lines 11-25 and page 25, line 1.**

“A cutoff of <1% positively stained cells was used to indicate ER/PR negativity in

immunohistochemistry testing,³⁰ and a HER2 status was defined negative by an
immunohistochemistry score of 0 or 1 or a lack of HER2 amplification (ratio < 2.2)
demonstrated by FISH analysis in accordance to the American Society of Clinical
Oncology/College of American Pathologists guideline.³¹ Ki67 was determined low if $\leq 14\%$
and high if $>14\%$ according to the St Gallen guidelines of 2013.³² According to ER, PR, HER2,
Ki67 status, luminal A subtype was defined by positive ER and PR, negative HER2 and low
Ki67; luminal B (HER2-) subtype was defined by positive ER or PR, negative HER2 and high
Ki67; luminal B (HER2+) subtype was defined by positive ER or PR, positive HER2 regardless
of Ki67 status; HER2 positive subtype was defined by negative ER and PR, positive HER2
regardless of Ki67 status; Triple negative subtype was defined by negative ER, PR and HER2
regardless of Ki67 status.”

**For “c”, perhaps go out to VAF 0.5 rather than 1.0 (it is very hard to compare these**
**VAFs, as they are quite small, and 0.5 would represent a CCF of 1.0 for a diploid tumor**
**with 100% purity).**

**Response:** In Figure 2c, according to the reviewer’s comments, we set the upper side to
0.5.

**For “d”, the Sankey plot is a little difficult to follow – could they just do stacked**
**barplots, with one bar for each of the tumor subtypes and then stacks of each of the 8**
**mutations? Could set these bars all to 1.0 total (same height), which would help us see the**
**distribution of mutations (as they already show the breakdown by subtype in (a)).**

**Response:** In Figure 2d, we apologize for the inconvenience. Compared with stacked
barplots, Sankey plots appear more abstract. However, in recent years, Sankey plots seem to be
widely used in diverse studies to represent arrow or arcs that have a width proportional to the
importance of the flow. Honestly, it is difficult to draw one bar for each of the tumor subtypes
and stacks of each of the 8 mutations when one patient carries more than one of the 8 mutations.
We think that it is applicable to draw one bar for each of the tumor subtypes and one bar for
each of the 8 mutations. However, we respect the reviewer’s thoughtful considerations and

suggestions. Therefore, we would like to retain the original Sankey plots and upload barplots
 in supplementary files.

Moreover, we analyzed all genomic mutations in addition to these top mutated genes and
 their association with different breast cancer subtypes. The results are presented in barplots and
 the following corresponding supplementary table (**Supplementary Table 4**). Therefore, we
 also observed *FAM47C* and *KDM6A* mutations were associated with triple negative subtype,
 *CBFB* mutations were associated with luminal A subtype and *XDH* mutations were associated
 with luminal B (HER2-) subtype.

Supplementary Table 4. Groupwise comparison of mutational frequency by molecular subtype related to Figure 2e.

Gene	Group 1	Group 2	Group 1		Group 2		P	FDR
			Total	Mutated	Total	Mutated		
TP53	Triple negative	Rest	206	164	928	441	< 0.001	< 0.001
TP53	HER2	Rest	206	153	928	452	< 0.001	< 0.001
PIK3CA	Luminal B (HER2-)	Rest	393	102	741	117	< 0.001	0.012
PTEN	Triple negative	Rest	206	21	928	28	< 0.001	0.012
AKT1	Luminal A	Rest	139	17	995	37	< 0.001	0.029
FAM47C	Triple negative	Rest	206	6	928	2	< 0.001	0.155
GATA3	Luminal B (HER2-)	Rest	393	52	741	54	< 0.001	0.176
CBFB	Luminal A	Rest	139	10	995	18	0.001	0.176
KDM6A	Triple negative	Rest	206	9	928	8	0.001	0.176
XDH	Luminal B (HER2-)	Rest	393	9	741	2	0.002	0.245

9

10

11

**References**

- 1. Scott AD, Huang KL, Weerasinghe A, Mashl RJ, Gao Q, Martins Rodrigues F, *et al.* CharGer:
clinical Characterization of Germline variants. *Bioinformatics (Oxford, England)* 2019, **35**(5):
865-867.
- 2. Lek M, Karczewski KJ, Minikel EV, Samocha KE, Banks E, Fennell T, *et al.* Analysis of
protein-coding genetic variation in 60,706 humans. *Nature* 2016, **536**(7616): 285-291.
- 3. Huang KL, Mashl RJ, Wu Y, Ritter DI, Wang J, Oh C, *et al.* Pathogenic Germline Variants in
10,389 Adult Cancers. *Cell* 2018, **173**(2): 355-370.e314.
- 4. Couch FJ, Shimelis H, Hu C, Hart SN, Polley EC, Na J, *et al.* Associations Between Cancer
Predisposition Testing Panel Genes and Breast Cancer. *JAMA oncology* 2017, **3**(9): 1190-
1196.
- 5. Palmer JR, Polley EC, Hu C, John EM, Haiman C, Hart SN, *et al.* Contribution of Germline
Predisposition Gene Mutations to Breast Cancer Risk in African American Women. *JNCI:*
*Journal of the National Cancer Institute* 2020.
- 6. Shen R, Seshan VE. FACETS: allele-specific copy number and clonal heterogeneity analysis
tool for high-throughput DNA sequencing. *Nucleic acids research* 2016, **44**(16): e131-e131.
- 7. Kandoth C, McLellan MD, Vandin F, Ye K, Niu B, Lu C, *et al.* Mutational landscape and
significance across 12 major cancer types. *Nature* 2013, **502**(7471): 333-339.
- 8. Razavi P, Chang MT, Xu G, Bandlamudi C, Ross DS, Vasan N, *et al.* The Genomic Landscape
of Endocrine-Resistant Advanced Breast Cancers. *Cancer cell* 2018, **34**(3): 427-438.e426.
- 9. Bertucci F, Ng CKY, Patsouris A, Droin N, Piscuoglio S, Carbuccia N, *et al.* Genomic
characterization of metastatic breast cancers. *Nature* 2019, **569**(7757): 560-564.
- 10. Angus L, Smid M, Wilting SM, van Riet J, Van Hoeck A, Nguyen L, *et al.* The genomic
landscape of metastatic breast cancer highlights changes in mutation and signature
frequencies. *Nature genetics* 2019, **51**(10): 1450-1458.
- 11. Yates LR, Knappskog S, Wedge D, Farmery JHR, Gonzalez S, Martincorena I, *et al.* Genomic
Evolution of Breast Cancer Metastasis and Relapse. *Cancer cell* 2017, **32**(2): 169-184.e167.
- 12. Leijen S, van Geel RM, Pavlick AC, Tibes R, Rosen L, Razak AR, *et al.* Phase I Study
Evaluating WEE1 Inhibitor AZD1775 As Monotherapy and in Combination With
Gemcitabine, Cisplatin, or Carboplatin in Patients With Advanced Solid Tumors. *Journal of*
*clinical oncology : official journal of the American Society of Clinical Oncology* 2016, **34**(36):
4371-4380.

13. Osman AA, Monroe MM, Ortega Alves MV, Patel AA, Katsonis P, Fitzgerald AL, *et al.* Wee-1
kinase inhibition overcomes cisplatin resistance associated with high-risk TP53 mutations in
head and neck cancer through mitotic arrest followed by senescence. *Mol Cancer Ther* 2015,
**14**(2): 608-619.
14. George S, Wang Q, Heinrich MC, Corless CL, Zhu M, Butrynski JE, *et al.* Efficacy and safety
of regorafenib in patients with metastatic and/or unresectable GI stromal tumor after failure of
imatinib and sunitinib: a multicenter phase II trial. *Journal of clinical oncology : official*
*journal of the American Society of Clinical Oncology* 2012, **30**(19): 2401-2407.
15. Sangai T, Akcakanat A, Chen H, Tarco E, Wu Y, Do KA, *et al.* Biomarkers of response to Akt
inhibitor MK-2206 in breast cancer. *Clinical cancer research : an official journal of the*
*American Association for Cancer Research* 2012, **18**(20): 5816-5828.
16. Dedes KJ, Wetterskog D, Mendes-Pereira AM, Natrajan R, Lambros MB, Geyer FC, *et al.*
PTEN deficiency in endometrioid endometrial adenocarcinomas predicts sensitivity to PARP
inhibitors. *Science translational medicine* 2010, **2**(53): 53ra75.
17. Razavi P, Dickler MN, Shah PD, Toy W, Brown DN, Won HH, *et al.* Alterations in PTEN and
ESR1 promote clinical resistance to alpelisib plus aromatase inhibitors. *Nature Cancer* 2020,
**1**(4): 382-393.
18. Loibl S, Gianni L. HER2-positive breast cancer. *Lancet (London, England)* 2017, **389**(10087):
2415-2429.
19. Swain SM, Baselga J, Kim SB, Ro J, Semiglazov V, Campone M, *et al.* Pertuzumab,
trastuzumab, and docetaxel in HER2-positive metastatic breast cancer. *The New England*
*journal of medicine* 2015, **372**(8): 724-734.
20. Verma S, Miles D, Gianni L, Krop IE, Welslau M, Baselga J, *et al.* Trastuzumab emtansine for
HER2-positive advanced breast cancer. *The New England journal of medicine* 2012, **367**(19):
1783-1791.
21. Geyer CE, Forster J, Lindquist D, Chan S, Romieu CG, Pienkowski T, *et al.* Lapatinib plus
capecitabine for HER2-positive advanced breast cancer. *The New England journal of medicine*
2006, **355**(26): 2733-2743.
22. Park JW, Liu MC, Yee D, Yau C, van 't Veer LJ, Symmans WF, *et al.* Adaptive Randomization
of Neratinib in Early Breast Cancer. *The New England journal of medicine* 2016, **375**(1): 11-
22.
23. Robson M, Im SA, Senkus E, Xu B, Domchek SM, Masuda N, *et al.* Olaparib for Metastatic
Breast Cancer in Patients with a Germline BRCA Mutation. *The New England journal of*

- *medicine* 2017, **377**(6): 523-533.
- 24. Hurvitz SA, Gonçalves A, Rugo HS, Lee KH, Fehrenbacher L, Mina LA, *et al.* Talazoparib in
Patients with a Germline BRCA-Mutated Advanced Breast Cancer: Detailed Safety Analyses
from the Phase III EMBRACA Trial. *The oncologist* 2019.
- 25. André F, Ciruelos E, Rubovszky G, Campone M, Loibl S, Rugo HS, *et al.* Alpelisib for
PIK3CA-Mutated, Hormone Receptor-Positive Advanced Breast Cancer. *The New England*
*journal of medicine* 2019, **380**(20): 1929-1940.
- 26. Kim SB, Dent R, Im SA, Espié M, Blau S, Tan AR, *et al.* Ipatasertib plus paclitaxel versus
placebo plus paclitaxel as first-line therapy for metastatic triple-negative breast cancer
(LOTUS): a multicentre, randomised, double-blind, placebo-controlled, phase 2 trial. *The*
*Lancet Oncology* 2017, **18**(10): 1360-1372.
- 27. Jones RH, Casbard A, Carucci M, Cox C, Butler R, Alehami F, *et al.* Fulvestrant plus
capivasertib versus placebo after relapse or progression on an aromatase inhibitor in
metastatic, oestrogen receptor-positive breast cancer (FAKTION): a multicentre, randomised,
controlled, phase 2 trial. *The Lancet Oncology* 2020, **21**(3): 345-357.
- 28. Yu H, Lee H, Herrmann A, Buettner R, Jove R. Revisiting STAT3 signalling in cancer: new
and unexpected biological functions. *Nature reviews Cancer* 2014, **14**(11): 736-746.
- 29. Sanchez-Vega F, Mina M, Armenia J, Chatila WK, Luna A, La KC, *et al.* Oncogenic Signaling
Pathways in The Cancer Genome Atlas. *Cell* 2018, **173**(2): 321-337.e310.
- 30. Hammond ME, Hayes DF, Dowsett M, Allred DC, Hagerty KL, Badve S, *et al.* American
Society of Clinical Oncology/College Of American Pathologists guideline recommendations
for immunohistochemical testing of estrogen and progesterone receptors in breast cancer.
*Journal of clinical oncology : official journal of the American Society of Clinical Oncology*
2010, **28**(16): 2784-2795.
- 31. Wolff AC, Hammond ME, Hicks DG, Dowsett M, McShane LM, Allison KH, *et al.*
Recommendations for human epidermal growth factor receptor 2 testing in breast cancer:
American Society of Clinical Oncology/College of American Pathologists clinical practice
guideline update. *Journal of clinical oncology : official journal of the American Society of*
*Clinical Oncology* 2013, **31**(31): 3997-4013.
- 32. Goldhirsch A, Winer EP, Coates AS, Gelber RD, Piccart-Gebhart M, Thürlimann B, *et al.*
Personalizing the treatment of women with early breast cancer: highlights of the St Gallen
International Expert Consensus on the Primary Therapy of Early Breast Cancer 2013. *Annals*
*of oncology : official journal of the European Society for Medical Oncology* 2013, **24**(9):
2206-2223.

Reviewers' Comments:

Reviewer #1:

Remarks to the Author:

I believe my concerns and comments have been addressed in the revised manuscript.

Reviewer #2:

Remarks to the Author:

The authors have addressed my comments. I am especially grateful for the addition of the supplementary tables comparing patient and tumor characteristics between Fudan, MSKCC, and TCGA; sites of biopsy in the three cohorts; and treatments given on the umbrella trial in the advanced-stage cohort.

My only comment is that the authors now state in the discussion that the increased frequency of TP53 and NF1 relative to MSKCC may be related to a reduced frequency of ER+/HER2- breast tumors in their cohort. But it seems they report in Figure 3b that TP53, NF1, and indeed AKT1 and KMT2C (2 additional genes associated with metastatic ER+/HER2- breast tumors as compared to primaries) were higher in frequency in their cohort than in MSKCC among ER+/HER2- tumors only. They could emphasize in the Discussion that the enrichment of these metastasis genes appears to be upheld in the ER+/HER2- subtype only, though I agree that differences in recruitment and resulting tumor characteristics could still contribute.

Point-by-point response to the reviewers' comments

Reviewer #1 (Remarks to the Author):

Comment: I believe my concerns and comments have been addressed in the revised manuscript.

Response: Thanks for the reviewer's kind comment.

Reviewer #2 (Remarks to the Author):

Comment: The authors have addressed my comments. I am especially grateful for the addition of the supplementary tables comparing patient and tumor characteristics between Fudan, MSKCC, and TCGA; sites of biopsy in the three cohorts; and treatments given on the umbrella trial in the advanced-stage cohort.

My only comment is that the authors now state in the discussion that the increased frequency of TP53 and NF1 relative to MSKCC may be related to a reduced frequency of HR+/HER2- breast tumors in their cohort. But it seems they report in Figure 3b that TP53, NF1, and indeed AKT1 and KMT2C (2 additional genes associated with metastatic HR+/HER2- breast tumors as compared to primaries) were higher in frequency in their cohort than in MSKCC among HR+/HER2- tumors only. They could emphasize in the Discussion that the enrichment of these metastasis genes appears to be upheld in the HR+/HER2- subtype only, though I agree that differences in recruitment and resulting tumor characteristics could still contribute.

Response: Thanks for the reviewer's insightful comment and valuable suggestion. As it is suggested, we add in the Discussion that the enrichment of *TP53* and *NF1* genes appears to be upheld in the metastatic HR+/HER2- subtype particularly. **Please find the corresponding change on Page 19, Line 10-12.**